# Data-driven prediction of continuous renal replacement therapy survival

Davina Zamanzadeh [1], Jeffrey Feng [2], Panayiotis Petousis [3], Arvind Vepa [2], Majid Sarrafzadeh[1], S. Ananth Karumanchi [4], Alex A. T. Bui [2,7] ✉ & Ira Kurtz [5,6,7]

Continuous renal replacement therapy (CRRT) is a form of dialysis prescribed to severely ill patients who cannot tolerate regular hemodialysis. However, as the patients are typically very ill to begin with, there is always uncertainty whether they will survive during or after CRRT treatment. Because of outcome uncertainty, a large percentage of patients treated with CRRT do not survive, utilizing scarce resources and raising false hope in patients and their families. To address these issues, we present a machine learning-based algorithm to predict short-term survival in patients being initiated on CRRT. We use information extracted from electronic health records from patients who were placed on CRRT at multiple institutions to train a model that predicts CRRT survival outcome; on a held-out test set, the model achieves an area under the receiver operating curve of 0.848 (CI = 0.822–0.870). Feature importance, error, and subgroup analyses provide insight into bias and relevant features for model prediction. Overall, we demonstrate the potential for predictive machine learning models to assist clinicians in alleviating the uncertainty of CRRT patient survival outcomes, with opportunities for future improvement through further data collection and advanced modeling.

Renal replacement therapy (RRT) encompasses a range of treatments that replace some capabilities of inadequately functioning kidneys[1]. Certain patients, typically because of hemodynamic compromise, are unable to tolerate hemodialysis, the most common form of RRT. These patients are instead considered for continuous renal replacement therapy (CRRT), which provides gentler treatment over a more prolonged period[2,3]. Despite several decades of use, there remains no widely-adopted consensus on clinical guidelines physicians should use to decide whether to initiate CRRT that will result in a good outcome[4–7]. The decision whether to put a patient on CRRT largely depends on the physician's assessment of the patient's medical history, vital signs, labs, and medications[8]. Unfortunately, it is estimated that approximately 50% of adults who are placed on CRRT do not survive[5,9–12], and as such, treatment with CRRT is often futile for both the patient and their families. Adding to this issue is the approach in tertiary and quaternary care centers to "pull out all the stops", resulting in the treatment of patients with CRRT as a last hope to survive[13]. Mitigating this uncertainty is important not only to ensure that CRRT is recommended to patients who will benefit from the treatment but also to discern those who will not. CRRT is a resource-intensive intervention (time, personnel, equipment, cost) and when used inappropriately is not in concert with the ideal approach to managing patients. A predictive tool to inform this clinical decision-making task can improve the number of positive patient outcomes, help optimize resource allocation, and provide support for clinicians to explain their decisions to the patients and their families.

[1]Department of Computer Science, University of California, Los Angeles, Los Angeles 90095 CA, USA. [2]Medical & Imaging Informatics Group, University of California, Los Angeles, Los Angeles 90095 CA, USA. [3]Clinical and Translation Science Institute, University of California, Los Angeles, Los Angeles 90095 CA, USA. [4]Department of Medicine, Cedars-Sinai Medical Center, Los Angeles 90048 CA, USA. [5]Department of Medicine, David Geffen School of Medicine, University of California, Los Angeles, Los Angeles 90095 CA, USA. [6]Brain Research Institute, David Geffen School of Medicine, University of California, Los Angeles, Los Angeles 90095 CA, USA. [7]These authors jointly supervised this work: Alex A. T. Bui, Ira Kurtz. ✉e-mail: buia@mii.ucla.edu

Here, we introduce a machine learning model on a target outcome of whether a patient should start CRRT. Unlike existing models that predict in-hospital mortality once CRRT has started[9,10,14–16], our model can inform the clinician whether CRRT should be initiated in the first place. We provide a unique and in-depth analysis of our predictive model and utilize a large, longitudinal dataset of patients who were placed on CRRT at University of California, Los Angeles (UCLA) and Cedars-Sinai Medical Center quaternary care hospitals in Los Angeles. We highlight the needed clinical parameters in several patient subgroups that should be monitored and evaluated before CRRT initiation to inform the key clinical decision whether a patient will survive CRRT treatment or not. Though the model as presented is not currently intended for clinical use, our work demonstrates its utility as incentive for external validation studies in the future.

## Results

### Machine learning for predicting patient outcomes on CRRT

We collected a dataset to develop a machine learning model to predict if a patient will survive after being placed on CRRT. This dataset consisted of three cohorts across two hospital systems for a total of $N = 8986$ patients. The UCLA: CRRT ($N = 4161$) and Cedars: CRRT ($N = 3263$) cohorts contained adult patients treated at the respective hospitals who were all placed on CRRT. The UCLA: Control cohort ($N = 1562$) contained adult patients treated at UCLA but were not placed on CRRT, matched to individuals in the UCLA: CRRT cohort (described in Methods "Data"). We combined four known outcomes (described in Methods "Data") to construct the binary outcome variable of whether a patient should be placed on CRRT (Fig. 1a). This unification resulted in relatively balanced distributions for both cohorts (UCLA: $N = 2241(53.9\%)$ should be placed on CRRT; Cedars Sinai: $N = 1801(55.2\%)$ should be placed on CRRT).

We associated each sample (patient-CRRT session pair) with electronic health record (EHR) data that we processed into various data tables (Fig. 1a; described in Methods "Data"). The features for modelling were collected from data within a predefined window of days before starting CRRT to accurately represent the clinical question of whether a patient will benefit from CRRT (Fig. 1c). After data preprocessing (described in Methods "Data Preprocessing"), we trained, tuned, and tested multiple predictive models (Fig. 1b; described in Methods "Model Hyperparameters and Tuning"). The train and validation splits consisted of patients who were on CRRT for a maximum of seven days, which captured the majority of the patient population (UCLA: $N = 2435(58.5\%)$; Cedars Sinai: $N = 2069(63.4\%)$). However, when applying the model to potential CRRT patients, it would not be possible to isolate use cases of the algorithm based on the number of days on CRRT (without another model). Therefore, we evaluated and reported results on subpopulations of patients in the test split consisting of patients who were on CRRT for up to seven days and conversely more than seven days.

### Evaluation of model performance

We investigated multiple experiments that consider different combinations of cohort data (described in Methods "Model Training and Evaluation"), with optimal models after tuning documented in Supplementary Table 1. We defined the most comprehensive model as the model we trained and evaluated on a combination of the UCLA: CRRT, Cedars: CRRT, and UCLA: Control cohorts, using only features shared across all cohorts (556 features). The stratified train, validation, and test splits consisted of 2999, 1015, and 991 samples, respectively. The inclusion of the control data distorted the proportion of outcomes in each split, with 41.0% recommended being put on CRRT. However, the isolated counts of UCLA: CRRT and Cedars: CRRT patients within the test split were more balanced at 529 (50.8% positive) and 340 (48.8% positive) patients, respectively. Classification performance on both the entire test split and cohort subgroups is illustrated in Fig. 2 (see blue curves). The model achieved a receiver operating characteristic area

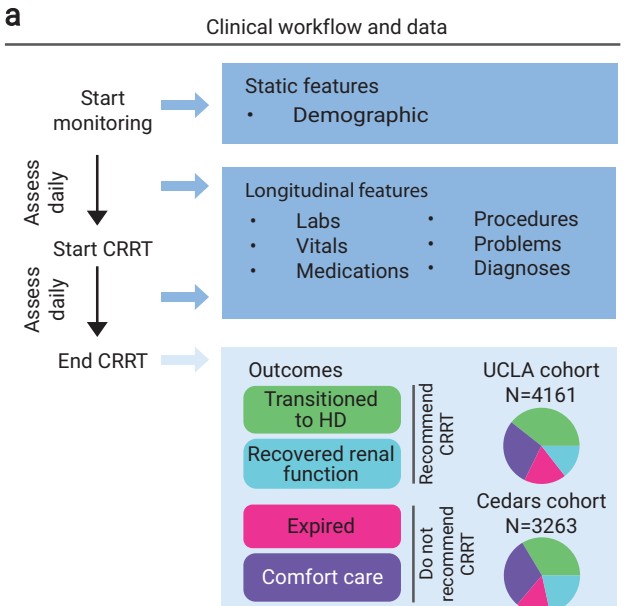

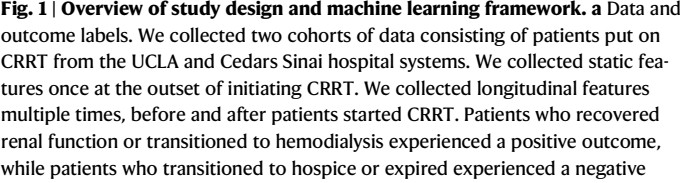

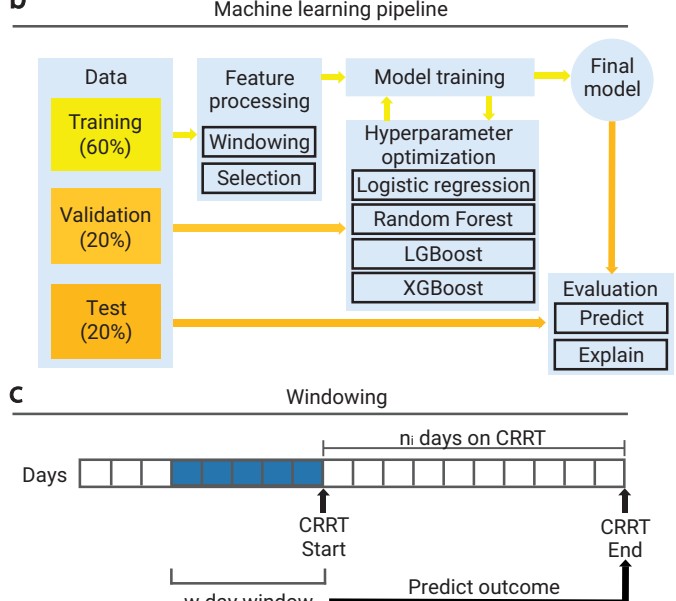

**Fig. 1 | Overview of study design and machine learning framework. a** Data and outcome labels. We collected two cohorts of data consisting of patients put on CRRT from the UCLA and Cedars Sinai hospital systems. We collected static features once at the outset of initiating CRRT. We collected longitudinal features multiple times, before and after patients started CRRT. Patients who recovered renal function or transitioned to hemodialysis experienced a positive outcome, while patients who transitioned to hospice or expired experienced a negative outcome. **b** Schematic of the machine learning pipeline. Training, tuning, and validation were performed on a 60/20% split of the dataset, with the remaining 20% as a holdout test set. External testing was performed on any unseen cohort. **c** Schematic of windowing of longitudinal features. Features were aggregated over a $w$ day window before the first day of CRRT. The features were used to predict the outcome, regardless of the number of days on CRRT.

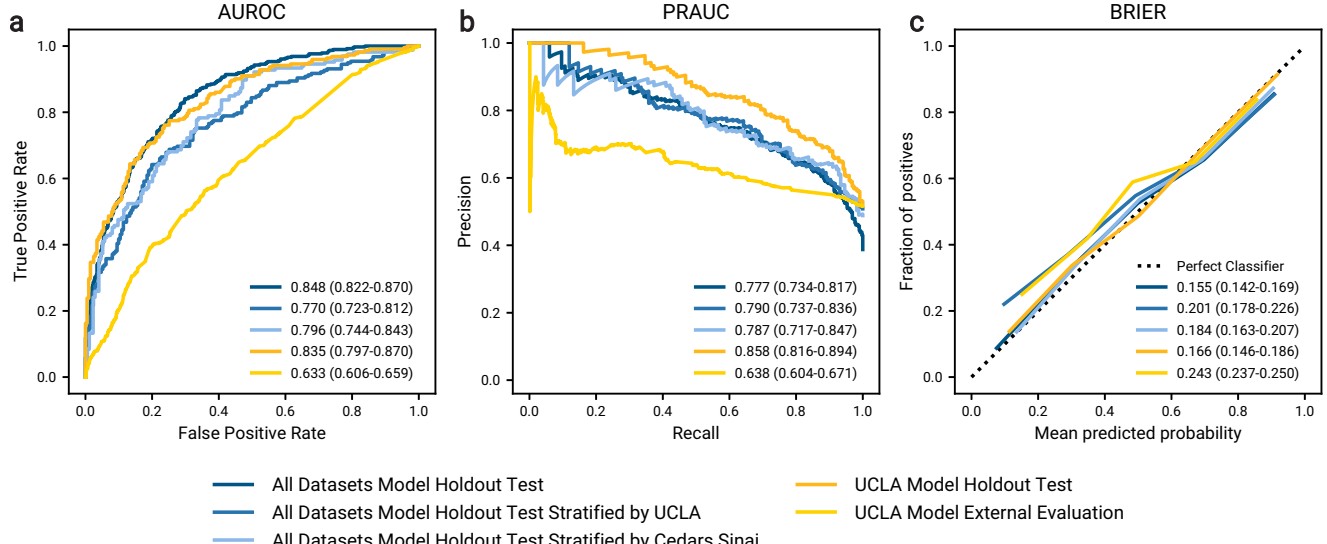

**Fig. 2 | Model performance when predicting CRRT patient outcomes.** Blue curves illustrate the performance of a model on the holdout test dataset ($N = 991$) after training on a combination of UCLA: CRRT, Cedars: CRRT, and UCLA: Control cohorts ($N = 2999$). The darkest blue curve illustrates the overall performance on the holdout test set, while the lighter and lightest blue curves illustrate the stratified results on the UCLA: CRRT ($N = 429$) and Cedars: CRRT ($N = 340$) constituents of the test dataset. Yellow curves illustrate the performance of a model trained on single-institution data from UCLA: CRRT ($N = 1268$), evaluated on both an internal holdout test dataset ($N = 425$) shown in darker yellow, and an external dataset from Cedars: CRRT ($N = 1788$) shown in lighter yellow. Reported statistics include point estimates as well as 95% confidence intervals obtained from 1000 bootstrap iterations of the test dataset. **a** Receiver operating characteristic curves for the prediction of CRRT outcome. Summarizing metric is the receiver operating characteristic area under the curve (ROCAUC). **b** Precision curves for the prediction of CRRT outcome. Summarizing metric is the precision recall area under the curve (PRAUC). **c** Calibration curves for the prediction of CRRT outcome. The summarizing metric is the Brier score. Source data are provided as a Source Data file.

under the curve (ROCAUC) of 0.848 (CI = 0.822–0.870) and precision recall area under the curve (PRAUC) of 0.777 (CI = 0.734–0.817), improving upon the uncertain CRRT outcomes observed in current clinical practice. The model was also well-calibrated, with a Brier score of 0.155 (CI = 0.142–0.169). Results of the isolated subgroups indicate comparable performance between the UCLA: CRRT cohort (ROCAUC: 0.770, CI = 0.723–0.812; PRAUC: 0.790, CI = 0.737–0.836) and the Cedars: CRRT cohort (ROCAUC: 0.796, CI = 0.744–0.843; PRAUC: 0.787, CI = 0.717–0.847).

## Model and feature interpretation

We evaluated the Shapley additive explanations (SHAP) values of the optimal model defined in Section "Evaluation of model performance" on the entire test split (Fig. 3a), as well as subsets based on a confusion matrix (Fig. 3b). The top ten features for all patients had significant overlap with the top ten features for the subset of patients that were true positives (TPs), true negatives (TNs), false positives (FPs), and false negatives (FNs) (Fig. 3b). The highlighted features included several diagnoses, conditions, and labs that were relevant for model prediction.

We also identified features that contributed to the majority of errors via model error analysis (Mealy)[17] and illustrated the path of the decision tree that contributes the most errors (Fig. 4a). In contrast with the features from the SHAP analysis, this analysis highlights different conditions, labs, and medications that contribute to prediction errors.

Lastly, we investigated if the feature distributions from type I and II errors were statistically different from the subset of correctly predicted patients that shared the same true label (Fig. 4b). The number of significantly different features between FN and FP patients from their correct counterparts was $N = 10$ and $N = 36$ respectively. Some of the features with the largest effect size were also important factors driving predictive decisions, suggesting that the model used the important feature values that distinguished errors from their correct counterparts

to discriminate between true and false predictions. These features are potential confounders that could use additional analyses.

## Clinical considerations of an applied model

Figure 5a visualizes the performance of the optimal model defined in Section "Evaluation of model performance". We reported the performance for different subgroups of patients (described in Methods "Subpopulation Analysis") from the holdout test set, as well as patients that were on CRRT for more than 7 days ($N = 3282$, 49.5% recommending CRRT). FN rates, FP rates, and confusion matrices for each subgroup are illustrated in Supplementary Fig. 1. Predictive performance was consistent for the patients with heart, liver, or infection indications, as well as those without any of these indications. We did not observe a difference in performance between sex or ethnicity groups. We observed a slightly poorer and greater variation of performance for Asian patients ($N = 258$ (8.1%)) and patients 90 to 100 years old ($N = 67$ (2.0%)), which is tied to the sample size of these groups. The trend between the balance of subgroup sample size and performance should be examined. Particularly, the causes of imbalance and explanations of potentially poor performance need to be explored.

When the model is used as a support tool in a decision-making process, the prediction threshold should be adjusted to take into consideration operational factors. We demonstrate outcomes under different operating thresholds in Fig. 5b. The current standard of clinical practice for patients considered for CRRT would be to always place patients on CRRT. As the threshold for placing patients on CRRT increases, fewer patients are recommended to be placed on CRRT. If the model did not recommend CRRT to patients who may have potentially benefitted from the treatment, then such FNs could result in increased mortality. Conversely, correctly not recommending treatment to those who would not benefit would save patient discomfort and resources. We will need to develop a procedure or

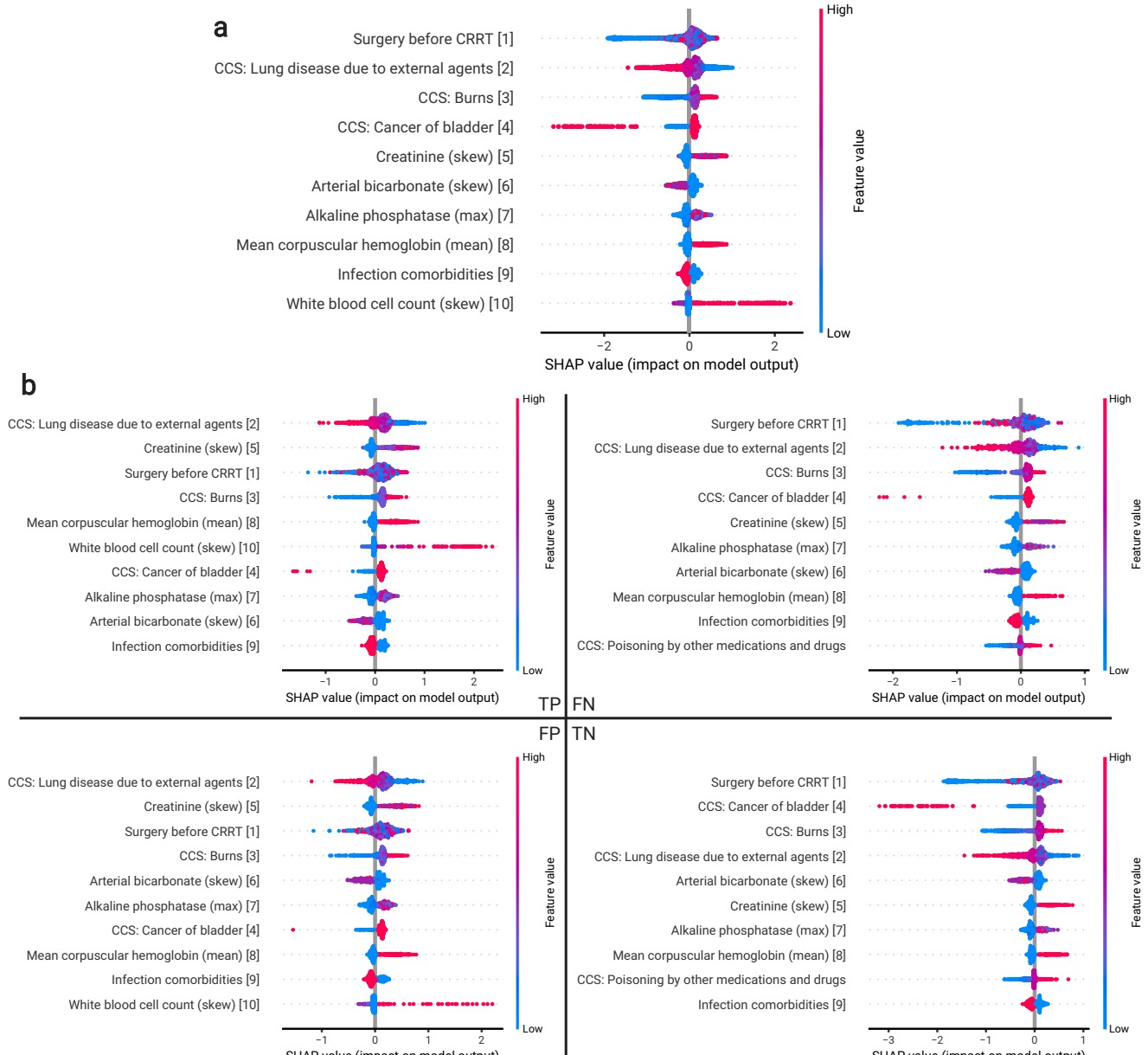

**Fig. 3 | Model explanations for the model trained and evaluated on a combination of the UCLA: CRRT, Cedars: CRRT, and UCLA: Control cohorts, using only features that exist across all three cohorts (defined in Methods "Evaluation of model performance").** SHAP values were evaluated using the holdout test set, in addition to patients that were on CRRT for more than 7 days (*N* = 3282 (49.5% recommending CRRT)). Red indicates that a higher feature value has the corresponding impact, as indicated by the x-axis, on model output. Blue indicates the impact of lower feature values on model output. **a** Ordered ranking of the ten most important features by average magnitude of SHAP values and direction of influence on output predictions. **b** Ordered ranking of the ten most important features by average magnitude of SHAP values when isolating subpopulations of patients in the test set that were classified as true positives, true negatives, false positives, and false negatives. We reported the corresponding rank of each feature in the ordered ranking of feature importance using the entire dataset for each feature (for the top ten features).

---

heuristic for determining the optimal operating point for any institution that would use our model in assisting with decision-making.

We also evaluated our model on a rolling basis (without retraining), on patients treated with CRRT for a maximum of seven days (Fig. 1c). This analysis allowed for the evaluation on data after patients started CRRT (which would change their biological processes), without leaking information into model training. We observed that the model improved when evaluating on the UCLA: CRRT stratum of the holdout test set as the data became more temporally close to, and therefore more relevant to, the patient's outcome. However, we did not observe a similar improvement on the Cedars: CRRT stratum of the holdout test set. These results suggest that the model learned meaningful relationships in the data before starting CRRT that maintained importance in the UCLA: CRRT cohort but not in the Cedars: CRRT cohort (Fig. 5c). We hypothesize that the decrease was due to relatively higher number of labs available in the UCLA: CRRT cohort compared to the Cedars: CRRT cohort. In general, the observations suggest that site-specific dynamic models that evaluate on a day-by-day basis may be more clinically relevant and a natural next step.

**Comparison between single and multi-site models**

We provide examples of externally validated local solutions via the cross-institutional application of a model trained on a single institution. Models were trained, validated, and tested on the patients and

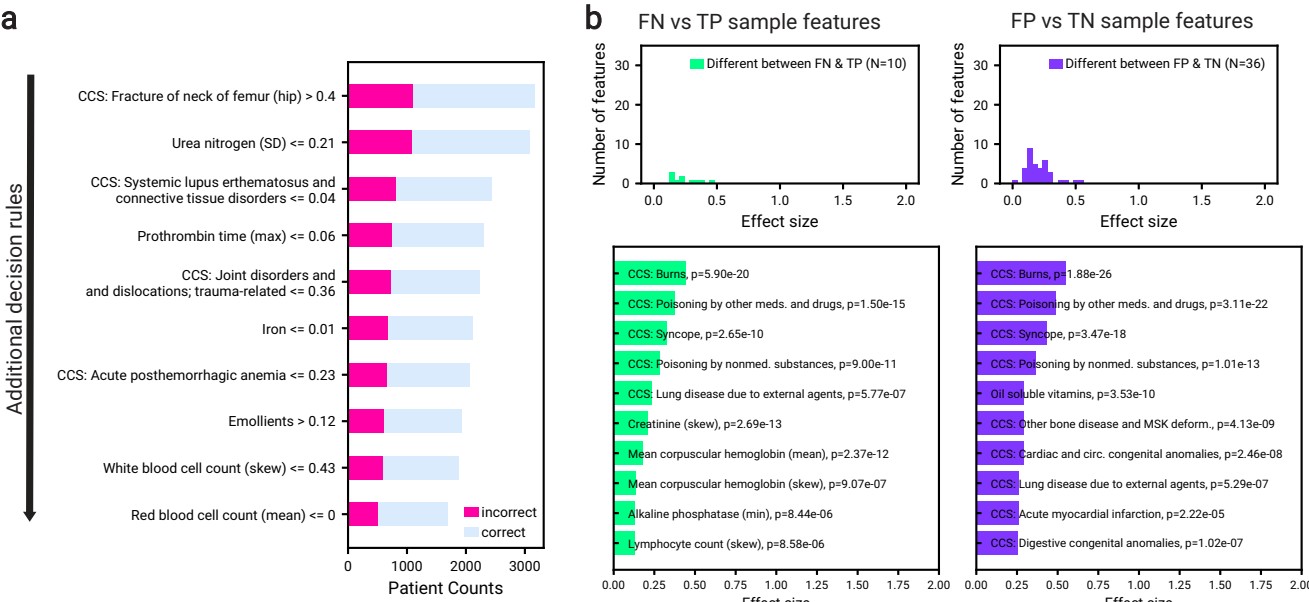

**Fig. 4 | Error analysis for the model trained and evaluated on a combination of the UCLA: CRRT, Cedars: CRRT, and UCLA: Control cohorts, using only features that existed across all three cohorts (defined in Section "Evaluation of model performance").** We performed analysis on the holdout test set, in addition to patients that were on CRRT for more than 7 days (N = 3282 (49.5% recommending CRRT)). **a** Top features that contributed to the majority of the errors. The threshold applied at each row operates on the resulting population after applying the respective threshold in the immediately above row. Pink indicates the number of incorrectly classified samples as a result of the decision threshold, while blue indicates the number of correctly classified samples. **b** Summary of analysis of model randomness against feature noise. Effect sizes are shown for the features that were significantly different between false negative and true positive populations (left, green). Effect sizes are also shown for the features that were significantly different between false positive and true negative populations (right, purple). Features with the top ten effect sizes are shown. Details of statistical test are described in 4.8. Source data are provided as a Source Data file.

features from a single cohort. The tuned models were then externally tested (with imputation of missing features) on all patients from another cohort. Figures 2 and 5c illustrate the performance of a model trained and validated on data from the UCLA: CRRT cohort (N = 1268 and N = 417 patient-CRRT session pairs respectively; both 51.7% positive), evaluated both on the internal test set (N = 425; 52.2% positive) and external Cedars: CRRT test set (N = 1788; 51.6% positive). The optimal model (Supplementary Table 1) demonstrated strong performance when applied to the internal test dataset (ROCAUC: 0.835, CI = 0.797–0.870; PRAUC: 0.858, CI = 0.816–0.894). However, the performance when externally testing on the Cedars: CRRT cohort decreased significantly (ROCAUC: 0.633, CI = 0.606–0.659; PRAUC: 0.638, CI = 0.604–0.671). Notably, we observe the opposite behavior when externally testing a model trained and validated on data from the Cedars: CRRT cohort (N = 1073 and N = 353 patient-CRRT session pairs respectively; both 51.1% positive), evaluated both on the internal test set (N = 366; 54.6% positive) and external UCLA: CRRT test set (N = 2149; 51.8% positive). Supplementary Fig. 2a demonstrates that the model trained on data from Cedars: CRRT cohort performed well when evaluating on data from the same institution, but decreased in performance when applied to the UCLA: CRRT cohort. On the other hand, both the comprehensive model in Section "Evaluation of model performance" and the model highlighted in Supplementary Fig. 2b were trained on data from multiple cohorts and exhibited better generalization when applied to data from multiple cohorts.

## Discussion

We present for the first time a machine learning model that predicts whether a patient should start CRRT. The current lack of consensus on clinical guidelines around CRRT initiation and pressure from family and consultants contribute to the often inappropriate use of CRRT. Scores such as APACHE II[18] and SOFA[19] provide baselines for ICU mortality and have been proposed for CRRT use. However, there is a

lack of specific models for solving issues regarding decision-making for CRRT initiation, and outcomes clearly remain suboptimal[9,14,15]. Ultimately, while many factors must be considered to determine the appropriateness of starting CRRT, we show that using EHR information in our model can help discern patients who will clinically benefit from this intervention, particularly within a 7-day timeframe. Our overall performance, as measured by ROCAUC and PRAUC, is a significant improvement over the current reported rates of only 37–55% of individuals surviving through this intervention[9–11].

Our model interpretation also provides an initial understanding of what features drive predictions and the nature of errors. For example, surgery before CRRT is a variable that predicts mortality (Fig. 3). This is consistent with published literature that CRRT in surgical patients have better outcomes than in medical patients[20,21]. Likewise, other factors in the model such as white blood cell count and creatinine have been linked with excess mortality in prior studies[20,22]. While some of the features are consistent with clinical intuition, we note that interpreting feature importance for this model is challenging due to the large number of potentially interacting variables. Future exploration of the role of key variables among other emergent features from our analyses and their causal connection to CRRT outcomes could contribute to objective clinical guidelines for CRRT initiation. As our ultimate goal is to create a parsimonious model and extract a more explainable subset of features for bedside use, in the future we will explore the trade-off between the data-driven discriminative features and clinically important and commonly measured ones.

Our model trained on multi-institutional data demonstrates the promising value of applying data-driven methods to CRRT. Yet, the labels and collected data may be biased by the individual medical provider and institution's protocol. We can understand generalizability of the model in more depth by evaluating with larger and more diverse datasets. On the other hand, even though generalizable models trained on data across multiple institutions are desirable for

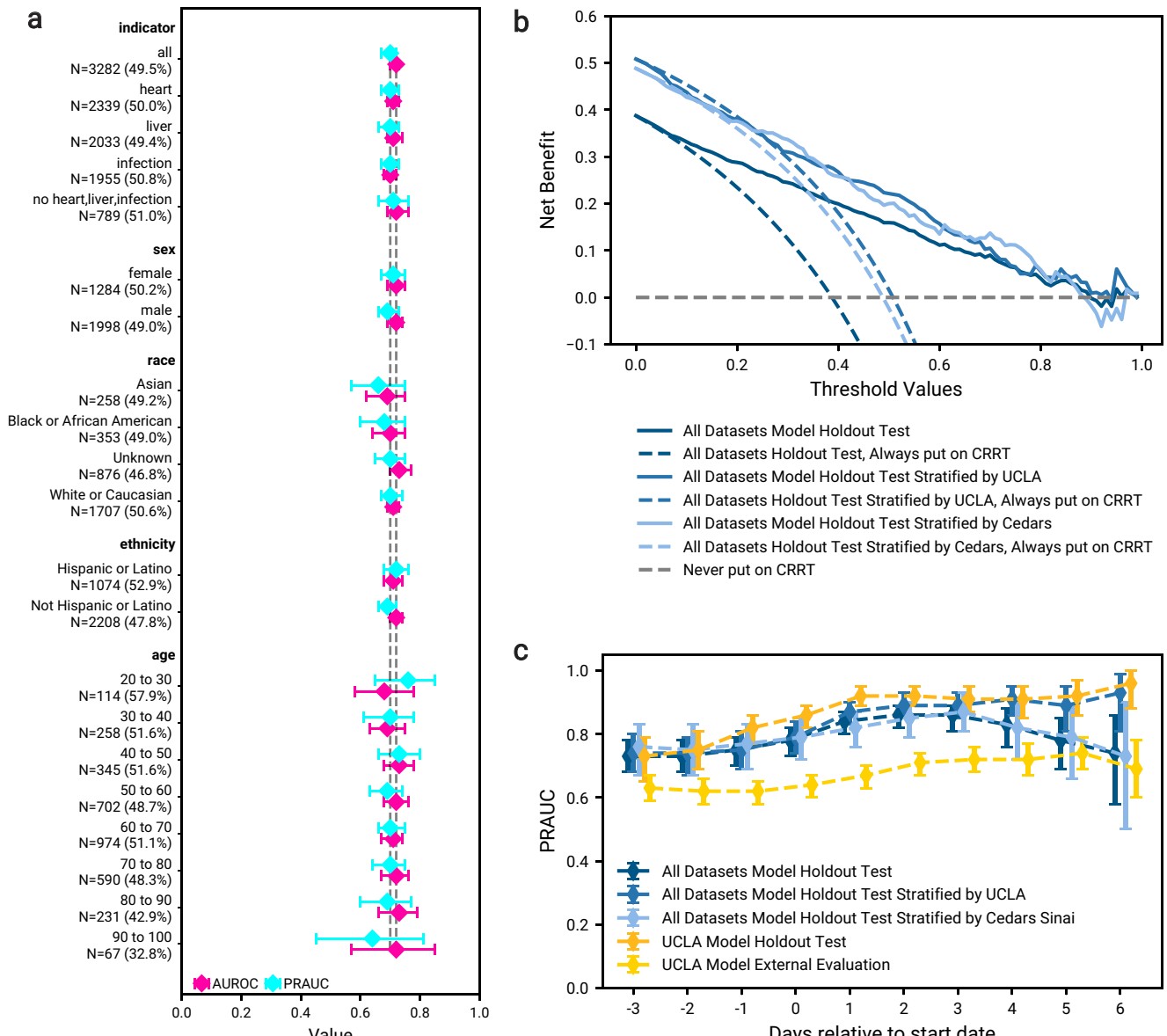

**Fig. 5 | Additional analyses and evaluation. a** Performance after training on a combination of UCLA: CRRT, Cedars: CRRT, and UCLA: Control cohorts (*N* = 2999), measured by ROCAUC (pink) and PRAUC (blue) on a holdout test set (*N* = 3282) including patients who were on CRRT for more than seven days (model defined in Section "Evaluation of model performance"). Results are also reported when applying the model to subgroups of the test set, categorized by disease indicator, sex, race, ethnicity, and age. Reported statistics include point estimates as well as 95% confidence intervals obtained from 1000 bootstrap iterations of the test dataset. **b** Decision curve analysis illustrating net benefit at different operating thresholds for the same models and datasets described in Fig. 2. **c** Evaluation of the same models and datasets described in 2, when using features from shifted windows relative to the start date. The observation window was shifted from three days before starting CRRT to six days after starting CRRT. The models were evaluated (without re-training) on the subset of test patients who had available features for each shifted window. Reported statistics include PRAUC as well as 95% confidence intervals obtained from 1000 bootstrap iterations of the test dataset. Source data are provided as a Source Data file.

knowledge discovery, we also show that local solutions may be adequate or even better than general solutions. However, this is only true if local solutions are deployed in appropriate contexts, especially if resources are constrained (e.g., limited access to training and evaluation data). Particularly, one aspect of an inappropriate localization would be to directly transfer another local model to a different institution. We observe decreases in performance due to domain shift which is potentially caused by discrepancies in data collection practices across institutions (e.g., higher number of labs available in the UCLA: CRRT cohort compared to the Cedars: CRRT cohort). Moreover, our analysis demonstrates the importance of incorporating control groups in the model development process, as they enable the model to

discern patterns distinguishing between mortality (while not on CRRT) and cases where patients do not benefit from CRRT.

We also highlight additional important considerations for using such a model in a more specific manner than a general calculator such as SOFA. Subgroup analysis provides insight into bias and the need for further assessment across important groups such as race and causes of inconsistent performance. We demonstrate a preliminary extension of the analysis to different ICU types (Supplementary Table 2), which reveals strong model performance on surgical ICU patients, while generally worse performance on medical ICU patients especially from the Cedars: CRRT cohort. Such findings present crucial considerations as we improve the models for robust translation. Additionally, our

model was unaware of possible changes that might occur after their treatment start date that might affect their outcome. This is because we could only use historical data of a patient before they began CRRT treatment to train the model. However, the rolling window analysis demonstrates that the model performance changed as the data used for inference got closer to each patient's endpoint. We can intuitively understand that the model made the best prediction it could at "day zero" with the available information, even though with updated data the model may have predicted differently. Further improvement may be discovered with a dynamic model. Moreover, we believe multiple models may be most helpful when used together to assist with this clinical decision-making task: deciding if a patient should initiate CRRT, predicting how many days of CRRT a patient would require to benefit from treatment, and finally if that patient should remain on CRRT on a day-by-day basis after initiating treatment.

The ability to choose a threshold is another aspect of tuning and granularity that our model provides over regular calculators. Due to natural variability in the decision-making process, we believe that tailoring to particular institutions and healthcare providers based on their needs could be beneficial. It is also important to analyze such sources of variability to provide insight into a more precise model. For example, comfort care patients who were only put on CRRT to keep the patient alive long enough for a family member to visit would still be considered a negative outcome in the current model. This is because our model and current dataset did not consider such alternative definitions of a positive outcome. A prospective detailed case analysis to compare decision-making side-by-side with our predictive model could further shed light on decision-making around CRRT initiation and improve this data-driven approach. Another informative case analysis would compare decision-making with and without model assistance.

Overall, this paper raises considerations for the future development of machine learning for CRRT, and the potential for further data collection to support additional evaluations. We demonstrate the utility of machine learning for CRRT and identify a space of features to consider. Next, we will take the steps necessary in order to achieve clinical translation, some of which we highlight. Simpler versions of this predictive model could make the model more easily understood by clinicians and more accessible to a wider range of institutions. For example, the model could be simplified by honing in on important features for different subgroups or site-specific features. Prospective clinical studies will be needed to test this model in real-world settings prior to clinical implementation.

## Methods
### Data
Our dataset consisted of three cohorts of retrospective data of routine clinical processes collected from two different hospital systems (UCLA and Cedars-Sinai Medical Center) with approval and waiver of consent from the Institutional Review Board (IRB) at UCLA (Protocol Number 19-00093). The first cohort was the UCLA: CRRT population ($N = 4161$ patients), which contained adult patients treated at UCLA that were all placed on CRRT between 2015 and 2021. Adult patients are those that are 21 years of age or older at the start date of treatment. The second was the UCLA: Control population ($N = 1562$ patients), which contained adult patients treated at UCLA between 2016 and 2021 who were not placed on CRRT. These non-CRRT patients were matched to individuals in the UCLA: CRRT cohort based on race, ethnicity, age, sex, and disease status (via the Charlson Comorbidity Index) via cosine similarity. CRRT patients were already identified prior to receiving the datasets, and therefore no ICD codes were used to categorize patients into the CRRT and UCLA: Control cohorts. The UCLA: Control population consisted of $N = 242$ (15.5%) patients with known in-hospital mortality. The last cohort was the Cedars: CRRT population ($N = 3263$ patients), which

contained adult patients treated at Cedars who received CRRT between 2016 and 2021.

Our outcome of interest was whether a patient should be placed on CRRT. For the CRRT cohorts at UCLA and Cedars, we constructed the final binary target from four clinical outcomes: recover renal function, transition to hemodialysis (inpatient), comfort care, and expired. The former two represented conditions in which we would recommend CRRT (a positive sample). We recommended CRRT if the patient's kidneys either completely recovered or the patient was able to stabilize on CRRT and could continue with hemodialysis. The latter two represented conditions in which we would not recommend CRRT (a negative sample). We did not recommend CRRT if the patient did not stabilize on CRRT and continued to end of life care or passed away while on treatment. All patients in the UCLA: Control cohort were not placed on CRRT and therefore did not have CRRT outcome labels. We therefore assigned these patients a negative outcome (to not recommend CRRT) based on the observation that they were not placed on CRRT. There is a possibility that the control cohort with known in-hospital mortality may have benefitted from CRRT, which may have been a source of noise in these assigned outcomes However, there were very few control patients that experienced mortality who consistently had elevated blood creatinine. Therefore, there were few control patients that would potentially fall under a different outcome.

Features describing demographics, vitals, medications, labs, medical problems, and procedures were available for all three cohorts. Demographics were collected once and consisted of information regarding age, sex, race, and ethnicity. Basic descriptive statistics (demographics, outcome variables) for all three cohorts are illustrated in Table 1. The remaining longitudinal features were collected at multiple time points, before and after starting CRRT. The diagnoses and medical problems were documented using ICD-10 codes, along with the respective dates of diagnosis and dates of entry. The procedures were documented using CPT codes and the dates of procedures. Vitals (e.g., temperature, weight, height, systolic/diastolic blood pressure, respiration rate, pulse oximetry, heart rate) were described via numeric value and observation time. Medications were recorded using pharmaceutical subclass identifiers, along with the order date. Lastly, labs were described with the name of the target component or specimen, the value of the result, and the order date.

### Data preprocessing
The electronic health record (EHR) data were processed for downstream use in machine-learning pipelines. We defined each sample as a (patient, CRRT session) pair with unique outcomes, as a given individual may have had more than one treatment. For samples in the UCLA: Control cohort, we constructed outcomes and randomly chose a procedure date from the set of procedure dates each patient had undergone to act as a proxy for a treatment start date. For each sample, we loaded and aggregated all longitudinal data over a predefined window of $d$ days. Furthermore, we filtered the training and validation samples to those who were on CRRT for a maximum of seven days. Training a model on patients on CRRT for longer periods of time (eight days or more) made it difficult for the model to learn meaningful relationships between the features before initiating CRRT (which was a pattern we saw with deeper analysis in Methods "Rolling Window Analysis"). These patients were distinct in that their clinical outcomes were temporally far from the clinical data used to train the model to make predictions. Thus, we found that seven days was the optimal time to retain enough patients to train a model and ensure that training data was recent in relation to patient outcome horizons to promote learning. The continuous features over the window were aggregated to the minimum, maximum, mean, standard deviation, skew, and number of measurements. The categorical features over the window were aggregated to the count of occurrences of each category. All ICD codes were converted to Clinical Classifications Software (CCS) codes in

**Table 1 | Description of cohort outcomes and demographics**

| | | UCLA CRRT | | | Cedars Sinai CRRT | | | UCLA Control |
|---|---|---|---|---|---|---|---|---|
| | | Overall | Do not recommend | Recommend | Overall | Do not recommend | Recommend | Do not recommend |
| n | | 4161 | 1920 | 2241 | 3263 | 1462 | 1801 | 1562 |
| Outcome, n (%) | | | | | | | | |
| | Comfort Care | 1179 (28.3) | 1179 (61.4) | - | 984 (30.2) | 984 (67.3) | - | - |
| | Expired | 741 (17.8) | 741 (38.6) | - | 478 (14.6) | 478 (32.7) | - | 242 (15.5) |
| | Recovered renal function | 602 (14.5) | - | 602 (26.9) | 704 (21.6) | - | 704 (39.1) | - |
| | Transitioned to hemodialysis | 1639 (39.4) | - | 1639 (73.1) | 1097 (33.6) | - | 1097 (60.9) | - |
| Total days on CRRT, median [Q1, Q3] | | 6.0 [3.0,12.0] | 5.0 [2.0,12.0] | 6.0 [3.0,12.0] | 5.0 [3.0,10.0] | 4.0 [2.0,9.0] | 6.0 [4.0,11.0] | 0.0 [0.0,0.0] |
| Year of CRRT, median [min, max] | | 2018 [2015,2021] | 2018 [2015,2021] | 2018 [2015,2021] | 2019 [2016,2021] | 2019 [2016,2021] | 2019 [2016,2021] | 2019 [2016,2021] |
| Number of previous CRRT Treatments, mean (SD) | | 0.2 (0.7) | 0.1 (0.5) | 0.3 (0.8) | 0.2 (0.5) | 0.1 (0.4) | 0.2 (0.5) | 0.0 (0.0) |
| Age, median [Q1, Q3] | | 60.0 [47.0,69.0] | 60.0 [48.0,69.0] | 59.0 [45.0,68.0] | 66.0 [56.0,75.0] | 68.0 [58.0,78.0] | 63.0 [53.0,73.0] | 61.0 [48.0,70.0] |
| Sex, n (%) | | | | | | | | |
| | Female | 1682 (40.4) | 765 (39.8) | 917 (40.9) | 1111 (36.3) | 537 (37.8) | 574 (35.0) | 627 (40.1) |
| | Male | 2479 (59.6) | 1155 (60.2) | 1324 (59.1) | 1946 (63.7) | 882 (62.2) | 1064 (65.0) | 935 (59.9) |
| Ethnicity, n (%) | | | | | | | | |
| | Hispanic or Latino | 1416 (34.0) | 580 (30.2) | 836 (37.3) | 756 (24.7) | 307 (21.6) | 449 (27.4) | 548 (35.1) |
| | Not Hispanic or Latino | 2745 (66.0) | 1340 (69.8) | 1405 (62.7) | 2301 (75.3) | 1112 (78.4) | 1189 (72.6) | 1014 (64.9) |
| Race, n (%) | | | | | | | | |
| | American Indian or Alaska Native | 15 (0.4) | 11 (0.6) | 4 (0.2) | 4 (0.1) | 1 (0.1) | 3 (0.2) | 30 (1.9) |
| | Asian | 337 (8.1) | 163 (8.5) | 174 (7.8) | 249 (8.1) | 103 (7.3) | 146 (8.9) | 131 (8.4) |
| | Black or African American | 429 (10.3) | 199 (10.4) | 230 (10.3) | 462 (15.1) | 226 (15.9) | 236 (14.4) | 158 (10.1) |
| | Multiple Races | 42 (1.0) | 15 (0.8) | 27 (1.2) | 115 (3.8) | 40 (2.8) | 75 (4.6) | 25 (1.6) |
| | Native Hawaiian or Other Pacific Islander | 21 (0.5) | 10 (0.5) | 11 (0.5) | 12 (0.4) | 3 (0.2) | 9 (0.5) | 9 (0.6) |
| | Unknown | 1268 (30.5) | 555 (28.9) | 713 (31.8) | 500 (16.4) | 287 (20.2) | 213 (13.0) | 591 (37.8) |
| | White or Caucasian | 2049 (49.2) | 967 (50.4) | 1082 (48.3) | 1715 (56.1) | 759 (53.5) | 956 (58.4) | 618 (39.6) |
| Heart comorbidities, n (%) | | | | | | | | |
| | No | 1115 (26.8) | 516 (26.9) | 599 (26.7) | 584 (17.9) | 189 (12.9) | 395 (21.9) | 1074 (68.8) |
| | Yes | 3046 (73.2) | 1404 (73.1) | 1642 (73.3) | 2679 (82.1) | 1273 (87.1) | 1406 (78.1) | 488 (31.2) |
| Liver comorbidities, n (%) | | | | | | | | |
| | No | 1159 (27.9) | 532 (27.7) | 627 (28.0) | 1353 (41.5) | 528 (36.1) | 825 (45.8) | 1220 (78.1) |
| | Yes | 3002 (72.1) | 1388 (72.3) | 1614 (72.0) | 1910 (58.5) | 934 (63.9) | 976 (54.2) | 342 (21.9) |
| Infection comorbidities, n (%) | | | | | | | | |
| | No | 1709 (41.1) | 821 (42.8) | 888 (39.6) | 925 (28.3) | 291 (19.9) | 634 (35.2) | 1334 (85.4) |
| | Yes | 2452 (58.9) | 1099 (57.2) | 1353 (60.4) | 2338 (71.7) | 1171 (80.1) | 1167 (64.8) | 228 (14.6) |
| Other comorbidities, n (%) | | | | | | | | |
| | No | 3925 (94.3) | 1831 (95.4) | 2094 (93.4) | 3142 (96.3) | 1423 (97.3) | 1719 (95.4) | 1315 (84.2) |
| | Yes | 236 (5.7) | 89 (4.6) | 147 (6.6) | 121 (3.7) | 39 (2.7) | 82 (4.6) | 247 (15.8) |

Age data was missing from 36 patients in the UCLA: CRRT cohort. Age, sex, ethnicity, and race data was missing from 206 patients in the Cedars: CRRT cohort. Note that comorbidities were identified if the CCS codes described in Section "Data Preprocessing" were recorded within 14 days before starting CRRT, which reflected the optimal window of the comprehensive model.

order to reduce the number of categories[23]. All procedure codes were converted to Current Procedural Terminology (CPT) codes[24,25]. We aligned the names of all vital, all medication, and most lab names across all cohorts in reference to the data contained in the UCLA: CRRT cohort. For vitals and labs, we also unified the units. Additionally, if any values were recorded as ranges bounded on one side, we assumed the value to be the bound (e.g., a value of > 4 was assumed to be 4). We then combined the outcomes, the aggregated longitudinal features, and the static features (patient demographic information). If any feature had a missingness rate of over 95%, we excluded it from our analysis.

We engineered a categorical feature from CCS codes that indicated common reasons for a patient requiring CRRT. Patients were categorized with an indication of heart problems (CCS codes: 96, 97, 100, 101, 102, 103, 104, 105, 106, 107, 108, 109, 114, 115, 116, 117), liver problems (CCS codes: 6, 16, 149, 150, 151, 222), severe infection (CCS codes: 1, 2, 3, 4, 5, 7, 8, 249), or some other disease indication.

These groups, aside from the "other" indication group were not mutually exclusive. Indications for heart problems occurred in $N = 3046$ (73.2%) of patients in the UCLA: CRRT cohort and $N = 2679$ (82.1%) in the Cedars: CRRT cohort. Indications for liver problems occurred in $N = 3002$ (72.1%) of patients in the UCLA: CRRT cohort and $N = 1910$ (58.5%) in the Cedars: CRRT cohort. Indications for severe infection occurred in $N = 2452$ (58.9%) of patients in the UCLA: CRRT cohort and $N = 2338$ (71.7%) in the Cedars: CRRT cohort. Lastly, in the UCLA: CRRT cohort $N = 326$ (5.7%) patients fell in the "other" category, while 94.3% fell in the former three groups. In the Cedars: CRRT cohort $N = 121$ (3.7%) fell in the "other" category, while 96.3% fell in the former three groups.

## Model training and evaluation

**Experiments.** We defined an experiment as the full procedure required to train and evaluate a unique predictive model. For each experiment, we specified a cohort or combination of cohorts on which we wanted to train and validate the model. The cohort(s) chosen were divided into train and validation splits using 60/20% of the data, respectively, with the remaining 20% as an internal test split. The entirety of any remaining cohort was used for external testing. As all samples in the UCLA: Control cohort had the same label, the control cohort test data was combined with the test data from the UCLA: CRRT cohort. Otherwise, the label homogeneity would have made it impossible to properly evaluate performance metrics. When training on multiple CRRT cohorts, performance was also evaluated on the isolated samples from each constituent CRRT cohort.

We explored all the following patterns to train and evaluate our methodology: 1) Experiment 1: train on UCLA: CRRT, evaluate on UCLA: CRRT, Cedars: CRRT, UCLA: Control; 2) Experiment 2: train on Cedars: CRRT, evaluate on Cedars: CRRT, UCLA: CRRT, UCLA: Control; 3) Experiment 3: train on UCLA: CRRT combined with Cedars: CRRT, evaluate on CRRT combined with Cedars: CRRT, UCLA: Control; and 4) Experiment 4: train on UCLA: CRRT combined with Cedars: CRRT combined with UCLA: Control, evaluate on CRRT combined with Cedars: CRRT combined with UCLA: Control. Experiments 1-4 were first conducted using the original number of features available in the training split. When training on multiple cohorts (Experiments 3 and 4), the total number of features comprised the union of the features of the individual cohorts. We also re-ran Experiments 1-4 with an extra feature selection step that reduced the feature set to those that existed in all three cohorts. Lastly, we re-ran Experiments 1-3 after reducing the feature set to those that existed in the UCLA: CRRT and Cedars: CRRT cohorts. While the main body of the paper focuses on Experiment 1 (using the original training features) and Experiment 4 (using the features that exist in all three cohorts), results for Experiment 2 (using the original training features) and Experiment 3 (using the features

that exist in the UCLA: CRRT and Cedars: CRRT cohorts) are illustrated in Supplementary Figs. 2, 3, 4, 5, 6, 7, 8.

**Model hyperparameters and tuning.** Each experiment yielded the optimal model of a grid of hyperparameters. Candidate models $m$ included linear and non-linear models: light gradient-boosting machine (LGBM), extreme gradient-boosted decision tree (XGB), random forest (RF), or logistic regression (LR). Each model type had its own (possibly different) hyperparameters that were also tuned as a part of the same grid. We selected the look-back window size $w$, or the number of days before each patient's treatment start date from which to aggregate data, from the options of 1, 2, 3, 4, 5, 6, 7, 10, and 14 days. We utilized simple imputation, which refers to mean imputation on continuous or quantitative features and mode imputation on categorical or qualitative features. Note that the imputation method was trained on the training split. Therefore, if features in the training cohort did not exist in the testing cohort, then entire features may have been imputed in the testing cohort. After imputation, the data were scaled between 0 and 1 using minimum-maximum scaling. We also decided on a feature selection procedure based on the Pearson correlation coefficient between each feature and the target variable. This selection was done by limiting to a particular number of features most correlated to the outcome ($k$-best, where $k \in \{5, 10, 15, 20, 25\}$) or by using a correlation threshold ($\rho$, where $\rho \in [0.01, 0.09]$ at intervals of 0.005). For a detailed breakdown of these hyperparameters, refer to Supplementary Algorithm 1. We ran $t = 400$ trials of tuning by randomly sampling from the above hyperparameter grid $t$ times. For each trial $t_j$, we loaded the data from the designated cohorts and divided it into the training, validation, and testing portions. We aggregated these three splits over the look-back window of $w_j$ days. The imputation and feature selection procedures were trained only on the training split but applied to all three splits. We trained the selected model $m_j$ on the train split and then evaluated its performance on the validation split. We then selected the model with the highest receiver operating characteristic area under the curve (ROCAUC) on the validation dataset and evaluated its performance on the testing dataset. Performance was measured by ROCAUC, PRAUC, Brier score, precision, recall, specificity, and F1 score. Confidence intervals were obtained through 1000 bootstrap iterations of the test split.

The optimal hyperparameters for all experiments after tuning are shown in Supplementary Table 1. Simple imputation and feature selection using a correlation threshold were optimal for all experiments. Supplementary Table 1 also describes the number of raw (before processing) and engineered (after processing) features that were available when using the optimal hyperparameters for each experiment. The feature counts are reported both before and after training. The features available for Experiments 1-4 were 2302, 1287, 2791, and 3220 respectively. When reducing the feature set to the intersection of the features between all three cohorts, the features available for Experiments 1–4 were 503, 556, 514, and 556, respectively. Lastly, when reducing the features to the intersection of the features between the UCLA: CRRT and Cedars: CRRT cohorts, the remaining features available for Experiments 1-3 were 626, 662, and 648 respectively.

## Subpopulation analysis

In addition to evaluating on the whole test dataset, we evaluated our model on subgroups of the dataset based on different characteristics. Evaluations in this analysis were performed on the combined set of patients who were on CRRT for up to seven days and more than seven days to reflect the performance of the model in a realistic setting. One set of characteristics included common medical reasons for requiring CRRT (identified via ICD code diagnoses), including heart issues, liver issues, and infection. These are some of the most common conditions in which a patient might be hemodynamically unstable and require

CRRT. For a list of codes we use for each indicator, refer to Section "Data Preprocessing". Note that these groups were not mutually exclusive; for instance, a patient may have had a heart condition and also be suffering from a severe infection. Other characteristics were based on sex, race, ethnicity, and age groups.

## Rolling window analysis

Each CRRT patient received treatment for a different number of days, so we may have made predictions on variable outcome horizons from the start of treatment. During that time, these patients were under active care, and their condition may have fluctuated due to their own physiologic processes or because of direct medical intervention. A CRRT patient's outcome may have also stemmed from events that occurred after they began treatment, meaning we would not be able to capture that data for new patients who had not been on CRRT. If the goal of our model is to predict if a patient should start CRRT, we could not train our model on post-start data which would leak information. Though the dynamic nature of CRRT posed a problem to training the model, we could evaluate our model on future data. We therefore implemented a rolling window analysis (visualized in Fig. 1a), evaluating the model (without retraining) on a set of $w$ day windows that each started a day later from the previous window. By monitoring metrics over each window, we could analyze how the predictions and the performance of the model changed as the data neared the outcome horizon (i.e., end of CRRT). If we observed an improvement in performance as the model was evaluated on data closer to the outcome horizon without retraining, we could infer that: 1) outcomes were influenced by events after they start CRRT; and 2) the model had learned meaningful information, and evaluating on the changing information had led to updated predictions for some patients on later windows. Similar conclusions could be drawn if we observed a decrease in performance as the model was evaluated on data further from the outcome horizon without retraining. We performed the rolling window analysis on patients on CRRT for a maximum of seven days because for patients with far-away outcome horizons, data after but near the start date might have still been uninformative.

Note that the rolling window analysis looked at contiguous days of data at once and evaluated a model trained on data before the start of treatment. This procedure was hence completely unaware of previous days or how measurements changed over time which would be used in an ongoing day-by-day predictive task.

## Explainability

It is critical when building machine learning models for clinical tasks that the model is understandable to promote transparency, correctness, and trust. We must understand why a model makes its decision, not just how and where it makes the most errors. To this end, in addition to technical performance metrics, we plotted the feature importance of the model applied to the entire test split via SHAP[26]. We also plotted feature importance for the subset of TP, FP, TN, and FN instances. Additionally, we visualized the features that contributed the most to error via Mealy. Mealy trains a secondary decision tree classifier to predict if the original model will output an incorrect or correct prediction[17].

To establish a deeper understanding of our model, we further evaluated if the model was learning distinctions between patients with incorrectly and correctly predicted outcomes. For each feature, we compared pairs of incorrectly and correctly predicted populations in the confusion matrix (i.e., FN vs. TP, FP vs. TN) and tested if the distributions of that feature were statistically significantly different between the two groups. Effect sizes complement the significant results to provide a measure of strength of difference, and thus highlight the features with values that differ the most between incorrectly classified samples and their correct counterparts. We employed a different statistical test and effect size formula depending on the type of feature. For a detailed breakdown of statistical tests and effect size formulas used, refer to

Supplementary Table 3. If we rejected, we reasoned that the model may have been using the feature to incorrectly distinguish the populations due to factors such as confounding, and such features with the largest effect sizes should be considered for additional analyses.

## Computational software

We implemented our procedure in Python 3.9.16. We primarily used the NumPy 1.23.0[27] and pandas 1.5.3[28] packages for loading and manipulating our datasets. Models were implemented via LightGBM 3.3.5[29] for an implementation of LGBM, xgboost 1.7.4[30] for an implementation of XGB, and Scikit-learn 1.2.2[31] for all other models. We used the Optuna 3.2.0[32] framework for model tuning and the algorithm for sampling from the hyperparameter grid. Statistical tests were computed using the SciPy 1.10.1 and Scikit-learn 1.2.2 packages. We used SHAP 0.42.0 for generating feature importance and explaining model decision-making on particular samples. Lastly, we used the Mealy 0.2.5 package for visualizing model errors.

## Statistical analysis

All statistical analyses were performed using Python 3.9.16 and SciPy 1.10.1. When comparing between-group differences in continuous variables, the Shapiro-Wilk test was first used to test for normality; then, normally distributed data were compared using the Student's $t$-test, and non-normally distributed data were compared using the Wilcoxon rank-sum test. Hedges' G statistic was used to describe the effect size. Binary-categorical variables were compared using Fisher's exact test, and Cohen's h was used to describe effect size. Multi-categorical variables were compared using the Chi-squared test, and Cramer's V was used to describe effect size. All statistical tests were two-sided and evaluated at a significance level of $p = 0.05$, with the Bonferroni correction for multiple testing.

## Reporting summary

Further information on research design is available in the Nature Portfolio Reporting Summary linked to this article.

# Data availability

We obtained approval from the UCLA IRB (Protocol Number 19-00093) to collect de-identified data from UCLA and Cedars Sinai Medical Center. UCLA IRB waived the need for consent on the use of retrospective data collected from routine clinical processes. As: (1) patients did not provide consent to participate in this study; and (2) the datasets used in this study are considered property of two health systems (UCLA, Cedars-Sinai), the data from this study can only be released to any third party by permission from our institutional Health Data Oversight Committee (HDOC). The policies and procedures that UCLA HDOC uses are based on revised guidelines put forth by the University of California (UC) on May 1, 2024. Requests for data releases can be made to Sandy Binder (SLBinder@mednet.ucla.edu), who is the UCLA HDOC Administrator. Requests will be evaluated against the five principles: (1) attention to the University's unique responsibility and mission; (2) sharing data outside UC for public benefit, (3) justice, (4) transparency and patient engagement, and (5) responsible stewardship. UCLA believes in a collaborative approach to enhancing reproducible science and encourages data sharing following its policies and procedures. HDOC reviews requests to ensure no apparent sale or barter of health data in exchange for goods, services, or other benefits; limit rediscDARlosure or reuse of health data with additional third parties without restriction; health data requests do not extend beyond the minimum necessary required to answer specific scientific hypotheses and that the request does not involve large volumes of data dealing with sensitive populations; sharing of potentially identifiable biometric data; conflicts of interest; and unusual or unique terms in the data use agreement that might pose risk to UC or patient population. All data releases require a data use agreement. HDOC meets biweekly and

requests can usually be reviewed within 2–4 weeks. Once HDOC approves a data release, UCLA's Technology Development Group will work to complete a data use agreement and provision the data. Source data are provided with this paper.

## Code availability

The corresponding code used in this work and examples are publicly available[33].

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

## Acknowledgements

We thank the UCLA Clinical and Translational Science Institute for their data collection support. This work was supported in part by funds from the Kidney, Urologic, and Hematologic-Advanced Research Training Program (NIH NIDDK U2C DK129496), the Medical Imaging Informatics Training Grant (NIH NIBIB T32 EB016640), and the UCLA CTSI grant (NIH UL1 TR001881). I.K. is supported in part by funds from the Smidt Family Foundation, the Factor Family Foundation, the Davita Allen Nissenson Research Fund, the Ralph Block Family Fund, and the Kleeman Family Fund.

## Author contributions

D.Z. and J.F. wrote the manuscript. D.Z., J.F., P.P., and A.V. contributed to data cleaning, model training, evaluation, and analysis of results. D.Z., J.F., P.P., A.V., M.S., and A.A.T.B. contributed to the overall design of the study, methods, and the manuscript. P.P., S.A.K., and I.K. acquired the data. I.K. and A.A.T.B. provided the original conceptualization of the study. All authors provided review, revision, and approval of the submitted manuscript.

## Competing interests

The authors declare no competing interests.
