## [Peer Review File · Nature Communications]

Data-driven prediction of continuous renal replacement therapy survivalREVIEWER COMMENTS

Reviewer #1 (Remarks to the Author):

The authors present a study titled "Data-driven prediction of continuous renal replacement therapy survival". The study includes the development of a machine learning algorithm to predict patient survival on CRRT. The goal of the model is to help inform and guide clinician decisions related to CRRT initiation. While the model is valuable and provides new insights into an understudied and complex population, there are significant revisions needed to clarify methods and results. There are a couple major comments including clarification on the suitability of the control group, translatability to a clinical setting and addressing limitations in the introduction and abstract. Specific comments by section are listed below.

General Comments

1. Within the text of the manuscript, the authors switch between past and present tense. It would be best to choose one to use throughout the entire document.
2. Throughout the manuscript there are run on sentences that need to be addressed.
3. Figures : Please switch all codes to feature names. Example Figure A7: CCS code 113 should be changed to the actual name.

Introduction

1. In the introduction the main purpose of the model is to inform clinicians. However, the results and discussion suggest that this is a "proof of principle" model and more data and advance modeling is needed before translation to the bedside. The last paragraph of the introduction should state that the model is not intended for clinical use at this time.

Methods

1. It is not specified why 7 days was chosen and only states that roughly half of CRRT patients fall within this time frame. Please add clarification on why this was chosen. Include discussion on whether the group of patients selected is homogenous when some patients will have much shorter CRRT duration.

2. Why was there no control group at Cedars? As shown in the results, having controls from only one institution introduces bias to the model. I know this would be a lot of work, but is it possible to obtain control individuals from this institution?

3. Please define TP, TN, FP and FN in the methods section. It is unclear if positive means that the patient would benefit or would not benefit from CRRT.

4. The control group line 659 is from a different time span than the CRRT groups. I am concerned that

A. ICD9 codes were used during this time. Were ICD9 codes equivalent to ICD10? This is not always the case.

B. Clinical practice changes over time and the controls from an older period of time may not be an adequate match to CRRT patients from a different time period. Would it be possible to run the model with controls for years similar to the CRRT population?

5. Line 672. The authors only looked at transition to hemodialysis. Peritoneal dialysis is another form that could be used after CRRT. Was this considered?

6. Line 681: control groups were categorized as should not be put on CRRT. Is it possible that some of them should have been to prevent mortality? Although no mortality was found in the control population. This seems unusual to me.

7. In the data section, please include ICD10 codes used and how CRRT patients were identified

8. In line 713 the authors created samples by pairing patient and a treatment. What is a treatment? Is this CRRT? Please specify

9. Did the model allow for the determination of interactions between features that were important for mortality risk in patients? It could be useful to examine interactions to see if the interaction between two minor features is as important as one of the top features.

10. In line 716: the authors state the control cohort the procedure date was randomly chosen. How was this done? Is there any way that this could have incorporated bias into the model.

11. Given the randomization is not clear and there is a lack of mortality, it is not clear that the control group is a good match for CRRT. It would be useful to have more clarity on the choice of control group (propensity score matching).

12. Lines 744 – 749. Why did the authors only choose three diagnoses groups. There are several others that are associated with CRRT patients. All diagnoses could be pooled into groups. If not done, the authors should state within the text what percentage of patients were in each of the diagnoses group including “other”.

13. In line 890, please include the percent of the population for each common medical condition associated with CRRT including “other”

Results

1. Line 268 – 274 It is unclear if the 10-day window includes before and after CRRT initiation. If it includes the 7 days while on CRRT this seems to negate the purpose of the model to inform whether CRRT should be used. Please clarify in the text.

2. Line 314 The results state that having previous treatments for CRRT “lends to a higher likelihood of recommending CRRT as it may indicate previous survival and benefit”. Was time between CRRT treatments included in the model as a confounding variable? Longer times between CRRT runs may indicate more stability, but if these treatments were given

within a short duration of time, it could indicate the patient is unstable.

3. Line 453 Add in cautionary statement that when using moderately strict threshold there will be some patients that the model decides should not be put on CRRT who should potentially be, and this could lead to mortality. The model is a useful tool, but it is a model and does come with caveats that should be stated along with the benefits.

4. Line 430-431 The model does “not perform well for older individuals”. What percent of the population on CRRT fall within the “older” category and what ages are included? If the largest part of the population is within this category, then that must be stated as it speaks to the translatability of the model.

5. In the results line 455 the authors say that “over 10,000 treatment days” would be saved using this model. There is no mention in the methods or results on how this was calculated. It is also unclear how this can be determined when the authors state the model needs more work before it can be used in a clinical setting.

6. In the results line 461 the authors say that the model will need to be optimized for each institution. Is there not a way to make the model generalizable based on features that are available in all institutions? How would you optimize to each institution? It does not seem like this is feasible.

7. Line 471 it is unclear what “more topical to patient’s outcome” is referring to. Please expand on this.

8. Line 498, It is unclear what the n numbers refer to, please add a description in the parentheses.

Discussion

1. In lines 542 – 549, The authors state that MCV could reflect the stability of the patient before CRRT or if a blood transfusion was given. While this is true, the severity of renal

dysfunction and CKD can also affect MCV. It would be helpful to include a discussion of the other important features, how they relate to the disease state and how this could affect survival. This would really speak to the value of the model for providing information on decision making.

2. Line 553: The authors state that removing important features that are not used at most institutions could be used in future models. This would create a more generalizable model. Would any of the most important model features be on this list? If so, could the model performance be affected?

3. Line 573-574: The authors state that “the control can discern between mortality (while not on CRRT) and cases where patients do not benefit from CRRT” The A1 table does not show any mortality for the control group. Again, this brings up the issue of the appropriateness of this control group for the CRRT patient group.

4. Line 621: I agree that comparing the model prediction with a clinician would be helpful, but would this need to be done at each institution? Again, this goes back to the feasibility of using this model in a clinical setting.

Reviewer #2 (Remarks to the Author):

In this article, Zamanzadeh et al introduce a machine-learning model focused on predicting whether a patient should initiate CRRT, providing a novel perspective compared to existing models that predict in-hospital mortality post-CRRT initiation. The model is based on a large, longitudinal dataset from the University of California, Los Angeles (UCLA) and Cedars-Sinai Medical Center. The aim is to improve clinical decision-making, optimize resource allocation, and enhance communication by offering valuable insights into the factors influencing the likelihood of positive outcomes with CRRT.

The machine learning pipeline consists of data feature processing, model training with hyperparameter optimization, and evaluation. The dataset is split into training (60%),

validation (20%), and test (20%) sets. The study employs logistic regression, Random Forest, LGBost, and XGBost models for prediction and explanation. The trained model is used to predict outcomes and provide explanations.

I found the article challenging to comprehend, particularly because clinically relevant variables often employed in such analyses, such as the delta change in creatinine, acidosis, pressors, FiO₂, PEEP, prior precipitating factors, co-morbidities, volume overloaded state (gain in weight), serum albumin, and others, were notably absent from the model.

Additionally, the practical utility of the created model for physicians at the bedside in the ICU remains unclear. Developing an algorithm (similar to the kidney failure risk equation), where variables can be inputted to estimate the risk of death over the subsequent 7 days, would enhance its usefulness. A risk of death > 90% can be considered futile. Currently, the clinical relevance of MCV measurements and tobramycin levels, as described, is uncertain.

The article, particularly in the discussion section, could benefit from a simpler presentation. It should include commentary on similar predictive models from prior research and explicitly highlight the distinctions from other models. Clarifying these points would contribute to a clearer and more accessible understanding of the study's findings.

Reviewer #3 (Remarks to the Author):

Well performed study and well written manuscript with overall high quality analysis. I have the following point by point comments:

1. most importantly: as the authors suggest, CRRT initiation is based on expert opinion. The authors created a matched population from the same hospital making it a single case control cohort with UCLA-CRRT as cases and UCLA-matched as controls. They matched patients according to demographics and clinical characteristics. This is similar to propensity score matching to reduce selection bias, but unfortunately treatment decision is very likely not random (treatment bias still exists). Can the authors define in this matched population why some patients were treated with CRRT and some are not? Comorbidity might be an

explanatory factor since this is higher in the "not recommended" cases vs controls.

2. line 680, it is not clear what the authors mean with patients should not be placed on CRRT and are labeled as such: isn't this a retrospective study where the patients are selected as controls as such?

3. line 703: based on what did the clinicians approve the fetare list? which criteria were used?

4. line 820: were features missing completely at random or were there dependencies? How did this influence imputation?

5. precision (in PRAUC) is dependent on prevalence (positive predictive value): were all patients in the respective hospitals included for analysis in the predefined time periods, i.e. is the prevalence correct?

6. How fair were the designed algorithms? How were race and ethnicity defined? It seems from fig A3 that the algorithms are somewhat underperforming for some of the minority groups.

7. There also seems to be worse performance in older patients.

8. four machine learning models were chosen, but it is unclear why these were selected, could the authors elaborate on this?

9. fig 2: what would be the effect of the overestimation at high probability and underestimation at low probability? Can the authors recalibrate perhaps?

10. perhaps I misread, but did the authors perform ensembling of the 4 machine learning models as well? or majority voting perhaps?

11. could the authors perform a net benefit analysis (decision curve analysis)? = for section 2.4, relatively similar to fig 5b

12. I'm not sure if fig 5b shows the right clinical scenario for potential implementation, because the fact that a patient is treated with CRRT is dependent on a previous choice, which suffers from treatment bias. What about patients not being placed on CRRT, but who would benefit? How does the 43% CRRT+poor outcome compare to other hospitals for instance? What would be the effect on the net benefit curve when this prevalence is changing (up or down)? Could the authors perhaps perform a simulation?

13. the decrease in model performance upon external evaluation is quite extensive: do the authors have an explanation (and perhaps also mitigation strategy) for this domain shift?

14. At the end, the model is for decision support, which indicates that the physician (+

patient) make the decision to initiate treatment and to continue or not. It might be worthwhile to perform a small crossover trial where physicians are randomized in either or not supported by the tool to see the impact on decision making.

Below, we respond to the comments provided (please note that the line numbers referred to in this document are in relation to the previously submitted version).

We would like to also note that in the process of updating this manuscript and rerunning the analyses, we have made the following changes to improve data harmonization and experimental efficiency:

- Converting ICD-9 to ICD-10 codes.
- Filtering all control patient data to be drawn from a similar period as the CRRT patients (2016-2021).
- Only use simple imputation.
- Doubling the number of trials for hyperparameter tuning.
- Increasing the range of k to try for k -best correlation feature selection.
- Removing features with a missingness rate of 95% or more.
- Better matching of labs and vitals between our UCLA and Cedars datasets.

Consequently, because of the modifications, we now have a smaller sample size, overall fewer features used in the model and some minor differences in the results. While we respond in our comments below to the reviews, we highlight a few additional changes here.

- The main performance metric in the abstract dropped (while still providing a high level of performance, there was a slight decrease).
- The optimal model for UCLA: CRRT improved when evaluating at UCLA on ROCAUC and PRAUC, while both dropped when externally validating on Cedars: CRRT data.
- We have a more similar performance between the UCLA: CRRT-only model and the Cedars: CRRT-only.
- There is improved model calibration.
- The top 10 features and the features the lead to the most error are different.
- There is less of a performance drop for certain minority groups and age groups.
- For the rolling window analysis of our Cedars Sinai cohort, there is a slight performance drop.

Again, we believe our responses and these changes address all the concerns raised in the review and significantly strengthen the overall impact of our work. We look forward to your consideration of this revised manuscript.

Reviewer 1

1. *Within the text of the manuscript, the authors switch between past and present tense. It would be best to choose one to use throughout the entire document.*

Response: We have revised the text to be consistent throughout.

2. *Throughout the manuscript there are run on sentences that need to be addressed.*

Response: This has been corrected as requested.

3. *Figures: Please switch all codes to feature names. Example Figure A7: CCS code 113 should be changed to the actual name.*

Response: This has been done as requested.

Introduction

1. *In the introduction the main purpose of the model is to inform clinicians. However, the results and discussion suggest that this is a “proof of principle” model and more data and advance modeling is needed before translation to the bedside. The last paragraph of the introduction should state that the model is not intended for clinical use at this time.*

Response: As requested, the following sentence has been added to the introduction, “Though the model as presented is not currently intended for clinical use, our work demonstrates its utility as incentive for clinical trials in the future.”

Methods

1. *It is not specified why 7 days was chosen and only states that roughly half of CRRT patients fall within this time frame. Please add clarification on why this was chosen. Include discussion on whether the group of patients selected is homogenous when some patients will have much shorter CRRT duration.*

Response: We have updated the text to add the following after lines 720-723: “Training a model on patients on CRRT for longer periods of time (eight days or more) made it difficult for the model to learn meaningful relationships between the features before initiating CRRT (which was a pattern we see with deeper analysis in Methods 4.5). These patients were distinct in that their clinical outcomes were temporally far from the clinical data used to train the model to make predictions. Thus, we found that seven days was the optimal time to retain enough patients to train a model and ensure that training data was recent in relation to patient outcome horizons to promote learning.”

For the model to operate appropriately, group homogeneity was not a factor or required. In addition, within the upper limit of seven days of possible treatment, we demonstrated predictive performance regardless of the number of days each individual patient was on CRRT. However, the group of patients excluded from training, (i.e., those with eight or more days on CRRT) have an important shared characteristic. Their inclusion causes the model performance to be less stable, likely due to the temporal problem mentioned above. We therefore picked a threshold based on a trade-off between model stability (based on how recent patient data would be in relation to the when the outcome occurred) and having enough patients to train the model.

2. *Why was there no control group at Cedars? As shown in the results, having controls from only one institution introduces bias to the model. I know this would be a lot of work, but is it possible to obtain control individuals from this institution?*

Response: We acknowledge that there is no control group at Cedars. Based on the analysis on data with and without the control group, including controls improves the performance under the AUROC and Brier Score in Figure 2. The PRAUC with the control data is poorer due to the imbalance in outcomes that adding the controls induces. We can expect that adding the control patients at Cedars would similarly improve the AUROC and Brier score while similarly compromising the PRAUC as the outcome imbalance becomes greater with their inclusion. That said, obtaining the control group data from Cedars is a lengthy process that would not allow us to resubmit our work in a timely manner.

3. *Please define TP, TN, FP and FN in the methods section. It is unclear if positive means that the patient would benefit or would not benefit from CRRT.*

Response: In the Methods section under the Data subsection, we modified the following sentences (changes underlined) to clarify what we define as positive outcome in lines 673-674 and 677-678: “The former two represented conditions in which we would recommend CRRT (a positive sample) ...The latter two represented conditions in which we would *not* recommend CRRT (a negative sample)...” This information is now conveyed before the ensuing subsections that mention TP, TN, FP, and FN for the first time in the Methods section.

4. *The control group line 659 is from a different time span than the CRRT groups. I am concerned that:*
 - a. *ICD9 codes were used during this time. Were ICD9 codes equivalent to ICD10? This is not always the case.*
 - b. *Clinical practice changes over time and the controls from an older period of time may not be an adequate match to CRRT patients from a different time period. Would it be possible to run the model with controls for years similar to the CRRT population?*

Response: We have made the following modifications:

- a. We made a conversion from ICD-9 to ICD-10 by using general equivalence mappings from the Centers for Medicare and Medicaid Services (2018 General Equivalence Mappings (GEMS) (ZIP)) and by the Department of Medicine at Columbia University Medical Centre (<https://github.com/AtlasCUMC/ICD10-ICD9-codes-conversion>).

b. We thank the reviewer for the suggestion. There is a trade-off between filtering patients to match in time and sample size for cohorts. When we include control patients from 2006-2022, there are N=2,364 patients (and as we sample one procedure date per patient, we also have 2,364 control samples). After filtering so that the control patients are also within the same time frame as the CRRT patients (2016-2022) there is a reduction to N=1,562. After this filtering, including fewer controls (which all have a “not recommend” outcome) increases the balance between negative and positive outcomes in the data. However, there is now a small degradation in performance of the model from AUROC from 87.5 to 84.8, and PRAUC from 79.8 to 77.7. However, we agree that ensuring that the control cohort is contemporaneous with the CRRT treated patients is preferable statistically. We have therefore as requested adjusted our analysis accordingly (2016-2022).

5. *Line 672. The authors only looked at transition to hemodialysis. Peritoneal dialysis is another form that could be used after CRRT. Was this considered?*

Response: Transition to hemodialysis following CRRT treatment refers specifically to in-hospital transition to hemodialysis that occurs when a patient is hemodynamically stable. At both institutions, it is very uncommon to transition to inpatient peritoneal dialysis. We therefore do not have a sample size to study transition to peritoneal dialysis in hospital from CRRT. We have now clarified in the text that the transition to dialysis from CRRT represents in-patient hemodialysis.

6. *Line 681: Control groups were categorized as should not be put on CRRT. Is it possible that some of them should have been to prevent mortality? Although no mortality was found in the control population. This seems unusual to me.*

Response: We thank the reviewer for the suggestion. There is in-hospital mortality recorded for the control population, which is 15.5% known in-hospital mortality for control patients between 2016-2022 (25.8% for all control patients). Of these 15.5% of cases, it is theoretically possible but not knowable that some patients should have been placed on CRRT. To address this question would require an in-depth case-by-case analysis that is beyond the scope of this study. However, in the revised manuscript, in lines 621-623 we propose that such an analysis be incorporated into future work. Based on this limitation, we acknowledge that some of the labels may be noisy. In addition, we have included mortality information by adding the following sentence after lines 656-652, which describe the control cohort: “The UCLA: Control population consisted of N=242 (15.5%) patients with known in-hospital mortality.” We further replaced lines 680-681 with the following sentences: “All patients in the UCLA: Control cohort were not placed on CRRT and therefore did not have CRRT outcome labels. We therefore assigned these patients a negative outcome (to not recommend CRRT) based on the observation that they were not placed on CRRT. There is a possibility that the control cohort with known in-hospital mortality may have benefitted from CRRT, which may have been a source of noise in these assigned outcomes. However, there were very few control patients that experienced mortality who consistently had elevated blood creatinine. Therefore, there were few control patients that would potentially fall under a different outcome.”

7. *In the data section, please include ICD-10 codes used and how CRRT patients were identified*

Response: There were no ICD codes used to identify patients for the CRRT cohorts. CRRT machines and consequently patient information from the cohort at both UCLA and Cedars Sinai are managed by a company called DaVita Inc. All outcome information was provided by DaVita. CRRT patients were de-identified by DaVita who performs the inpatient dialysis at UCLA and Cedars and keeps record of every patient who received inpatient CRRT. In lines 890-892, we clarify that ICD-10 codes were only used to group patients based on indications of heart issues, liver issues, and infection groups.

8. *In line 713 the authors created samples by pairing patient and a treatment. What is a treatment? Is this CRRT? Please specify.*

Response: To make this clearer we have changed the phrase *treatment* to *CRRT session* in this context. For example: “(patient, treatment) pair” changes to “(patient, CRRT session) pair.”

9. *Did the model allow for the determination of interactions between features that were important for mortality risk in patients? It could be useful to examine interactions to see if the interaction between two minor features is as important as one of the top features.*

Response: We agree that feature interaction is interesting and would be a potentially illuminating analysis. However, there are a few considerations that have led us to conclude that it would not be appropriate to pursue this avenue currently. The mathematical approaches available that address feature interaction can only test for the existence of an interaction, but not magnitude. Importantly, it would therefore not be possible to compare feature interaction with feature importance i.e., whether two minor features are as important as a single major feature. Moreover, it would be extremely complicated in our current study to analyze the interactions amongst pairs from the approximately 200 features included in the model. In future work, the magnitude of all potential feature interactions would be an analysis that we can attempt with a parsimonious model that includes fewer but more succinct features.

10. *In line 716: the authors state the control cohort the procedure date was randomly chosen. How was this done? Is there any way that this could have incorporated bias into the model.*

Response: We randomly chose a procedure date by randomly sampling one existing procedure for each patient and selecting the date. We have amended the text to make this clearer: “For samples in the UCLA: Control cohort, we constructed outcomes and randomly chose a procedure date from the set of procedure dates each patient had undergone to act as a proxy for a treatment start date.” Our reasoning is that a procedure, like a treatment, has a date and surrounding clinical events (i.e., diagnoses, prescriptions, etc.). The question of bias given this clarification is a complex one, and there are trade-offs between different types of bias. One question of bias could be how the distribution of procedure dates chosen compares to the CRRT population. We believe that it is not necessary or beneficial for the distribution of procedure dates to exactly match those of the CRRT population treatment dates. Bias can also appear with temporal considerations, where the control cohort and treated group timeframes differ (addressed in earlier previous response). Another concern could be the volume of data surrounding this anchoring date (the procedure date selected). For example, one could over-represent “sicker” patients with more labs and vitals drawn around their randomly chosen procedure date or over-represent “healthier” patients with less clinical data. The separate additional concern of random sampling is that the sample produced may happen to be sampled badly (i.e., in a skewed or incommensurate manner). Given the reviewer’s important comments, we analyzed the frequency of procedure dates selected over the study timeframe and did not find any skew patterns. Thus, we believe that our method of selecting a procedure date produces varying levels of volume of data that match what normally occurs in the population and most limits potential sources of bias.

11. *Given the randomization is not clear and there is a lack of mortality, it is not clear that the control group is a good match for CRRT. It would be useful to have more clarity on the choice of control group (propensity score matching).*

Response: We have addressed this concern in our earlier response and in the revised text. In lines 656-663, we elaborate on the matching algorithm used where we sample from the non-CRRT population at UCLA matching patients by age, sex, race, and ethnicity. Subsequently, we compute the Charlson Comorbidity Index for UCLA: CRRT and control cases. Based on that score, we match patients 1:1 using cosine similarity. To compute the Charlson Comorbidity Index, we use the equation defined at the following site <http://mchp-appserv.cpe.umanitoba.ca/viewConcept.php?conceptID=1098#3DIGITCODE>. Therefore, the control population are sick patients treated at UCLA for any reason and were not placed on CRRT. The control patients are similar to those who did go on CRRT in regard to demographics and disease status.

12. *Lines 744 – 749. Why did the authors only choose three diagnoses groups. There are several others that are associated with CRRT patients. All diagnoses could be pooled into groups. If not done, the authors should state within the text what percentage of patients were in each of the diagnoses group including “other.”*

Response: These three diagnoses groups were selected as they account for around 95% of cases that received CRRT at both institutions. We have amended the text to include the breakdown in addition to Table A1,

especially delineating how small the “other” group is. The text now reads (underline): “Patients were categorized with an indication...or some other disease indication.” We have also added the following information in-text in the following paragraph: “These groups, aside from the “other” indication group were not mutually exclusive. Indications for heart problems occurred in N = 3,046 (73.2%) of patients in the UCLA: CRRT cohort and N = 2,679 (82.1%) in the Cedars: CRRT cohort. Indications for liver problems occurred in N = 3,002 (72.1%) of patients in the UCLA: CRRT cohort and N = 1,910 (58.5%) in the Cedars: CRRT cohort. Indications for seiver infection occurred in N = 2,452 (58.9%) of patients in the UCLA: CRRT cohort and N = 2,338 (71.7%) in the Cedars: CRRT cohort. Lastly, in the UCLA: CRRT cohort N = 326 (5.7%) patients fell in the “other” category, while 94.3% fell in the former three groups. In the Cedars: CRRT cohort N = 121 (3.7%) fell in the “other” category, while 96.3% fell in the former three groups.”

13. *In line 890, please include the percent of the population for each common medical condition associated with CRRT including “other.”*

Response: We acknowledge there are other minor disease indications for CRRT. However, these represent only 5% of our total cohort. We therefore have labeled this group as “Other.”

Results

1. *Line 268 – 274 It is unclear if the 10-day window includes before and after CRRT initiation. If it includes the 7 days while on CRRT this seems to negate the purpose of the model to inform whether CRRT should be used. Please clarify in the text.*

Response: The window only includes data before CRRT initiation. To improve readability in text, we remove mention of optimal parameters, and instead refer to Table A2 that summarizes the optimal hyperparameters of each experiment. Table A2 clarifies that the window refers to “window before CRRT initiation.”

2. *Line 314 The results state that having previous treatments for CRRT “lends to a higher likelihood of recommending CRRT as it may indicate previous survival and benefit”. Was time between CRRT treatments included in the model as a confounding variable? Longer times between CRRT runs may indicate more stability, but if these treatments were given within a short duration of time, it could indicate the patient is unstable.*

Response: We thank the reviewer for the comment. In our most recent analysis, the model no longer considers the number of treatments of CRRT as one of top 10 most important features. We therefore removed this statement. Still, had it remained important, most patients in this study only received one CRRT treatment course. Considering the time between treatments as a feature would have limited utility and would be difficult to define for most of the population. These considerations could be potentially subject to a future study specifically addressing patients who received multiple treatment courses at separate times.

3. *Line 453 Add in cautionary statement that when using moderately strict threshold there will be some patients that the model decides should not be put on CRRT who should potentially be, and this could lead to mortality. The model is a useful tool, but it is a model and does come with caveats that should stated along with the benefits.*

Response: As requested, we re-emphasized the trade-off between increasing true negatives, which is cost/resource saving at the consequence of potential false negatives, which is life-threatening. Accordingly, we have modified lines 449-455: “If the model did not recommend CRRT to patients who may have potentially benefitted from the treatment, then such false negatives could result in increased mortality. Conversely, correctly not recommending treatment to those who would not benefit would save patient discomfort and resources.” We believe that the current model would need to be evaluated in a random prospective trial that is institution-specific to address its utility clinically and the optimal thresholds to use.

4. *Line 430-431 The model does “not perform well for older individuals”. What percent of the population on CRRT fall within the “older” category and what ages are included? If the largest part of the population is within this category, then that must be stated as it speaks to the translatability of the model.*

Response: We have changed the text to be more specific and to explicitly define the years we were referring to

by the term “older” patients. The data for these groups are found in Figure 5a. We have added their data and relative percentages to the split size in-text to convey that these groups are small. The new sentence is: “We observed a slightly poorer and greater variation of performance ... patients 90 to 100 years old (N = 67 (2.0%)), which is tied to the sample size of these groups.”

5. *In the results line 455 the authors say that “over 10,000 treatment days” would be saved using this model. There is no mention in the methods or results on how this was calculated. It is also unclear how this can be determined when the authors state the model needs more work before it can be used in a clinical setting.*

Response: As with any model of this type, as stated, the current model would need to be evaluated in a random prospective trial that is institution-specific to address its utility clinically. Accordingly, we have replaced the corresponding graph with a decision curve and have removed any discussion/analysis of days saved from the text and its corresponding graph in Figure 5.

6. *In the results line 461 the authors say that the model will need to be optimized for each institution. Is there not a way to make the model generalizable based on features that are available in all institutions? How would you optimize to each institution? It does not seem like this is feasible.*

Response: For any domain that uses machine learning, it is currently common practice to fine-tune models for their specific use case. To optimize the model for a given institution, one would train on a small set of data that is institution specific. It is unusual to build a single generalized model that outperforms a fine-tuned one, given the possibility of local confounders/influences. The cost of fine-tuning is not expensive, especially compared to potential benefits and the relative cost of other institutions having to start from scratch.

7. *Line 471 it is unclear what “more topical to patient’s outcome” is referring to. Please expand on this.*

Response: We have altered the text to make it clearer that “topical” means relevance based on nearness in time. The new sentence reads: “We observed that the model improved...as the data became more temporally close to, and therefore more relevant to, the patient’s outcome.”

8. *Line 498, It is unclear what the n numbers refer to, please add a description in the parentheses.*

Response: We have added to the text that N refers to patient-CRRT session pairs. The sentence now reads: “Figure 2 illustrates the performance of a model trained and validated on data from the UCLA: CRRT cohort (N=1,270 and N=416 patient-CRRT session pairs respectively; both 52.6% positive), evaluated both on the internal test set (N=430; 52.6% positive) and external Cedars: CRRT test set (N=1,790; 51.6% positive).”

Discussion

1. *In lines 542 – 549, The authors state that MCV could reflect the stability of the patient before CRRT or if a blood transfusion was given. While this is true, the severity of renal dysfunction and CKD can also affect MCV. It would be helpful to include a discussion of the other important features, how they relate to the disease state and how this could affect survival. This would really speak to the value of the model for providing information on decision making.*

Response: We agree with clinically interpreting important features. We added the following discussion after lines 541-543: “For example, surgery before CRRT is a variable that predicts mortality (Fig. 3). This is consistent with published literature that CRRT in surgical patients have better outcomes than in medical patients [20, 21]. Likewise, other factors in the model such as white blood cell count and creatinine have been linked with excess mortality in prior studies [20, 22].”

Conversely, we also acknowledge the challenge of interpreting the clinical relevance of features in our model, especially due to the large number of variables that can contribute and interact in multiple unknown ways that need to be understood in a future study. We also add the following to the text after the new lines of text: “While some of the features are consistent with clinical intuition, we note that interpreting feature importance for this model is challenging due to the large number of potentially interacting variables.” The top 10 most important features listed (Fig. 3) only partially convey why the model makes its decisions, but do not at all capture the complex (non)linear interactions of hundreds of available features the model considered. A deep understanding

of the features and their (non)linear interactions as addressed in a previous response is a separate undertaking that would be undertaken in a future study.

2. *Line 553: The authors state that removing important features that are not used at most institutions could be used in future models. This would create a more generalizable model. Would any of the most important model features be on this list? If so, could the model performance be affected?*

Response: It is theoretically possible that one can remove a given feature that is highly discriminative, however this could affect model performance as a tradeoff. The trade-off exists between data-driven discriminative features and clinically important and commonly measured features. Accordingly, we have changed the text: “As our ultimate goal is to create a parsimonious model and extract a more explainable subset of features for bedside use, in the future will explore the trade-off between the data-driven discriminative features and clinically important and commonly measured ones.” Determining which if any of these features to remove from a future model that has ultimate utility clinically should be done by empirically testing the inclusion/removal of specific highly discriminative features in a randomized prospective fashion.

3. *Line 573-574: The authors state that “the control can discern between mortality (while not on CRRT) and cases where patients do not benefit from CRRT” The A1 table does not show any mortality for the control group. Again, this brings up the issue of the appropriateness of this control group for the CRRT patient group.*

Response: We thank the reviewer for the comment. We have accordingly included mortality information in-text (outlined in our response to Method 6), as well as updated Table A1 to include this information.

4. *Line 621: I agree that comparing the model prediction with a clinician would be helpful, but would this need to be done at each institution? Again, this goes back to the feasibility of using this model in a clinical setting.*

Response: It is not necessary to compare this general “base” model to clinicians at each institution, only fine-tuning the model would be recommended to improve the model performance locally. Comparing this base model’s predictions to a group of clinicians is only necessary to broadly assess the behavior of the model relative to clinicians.

Reviewer 2

1. *I found the article challenging to comprehend, particularly because clinically relevant variables often employed in such analyses, such as the delta change in creatinine, acidosis, pressors, FiO2, PEEP, prior precipitating factors, co-morbidities, volume overloaded state (gain in weight), serum albumin, and others, were notably absent from the model.*

Response: The model uses all data in the EHR (excluding variables missing more than 95% data across all patient-CRRT session pairs). All available labs, clinical information, diagnoses, procedures, medications, etc., including those listed by the reviewer, are considered by the model. However, not all the features analyzed fall within the ten most important features according to the model. Nonetheless, after rerunning our analysis, several variables such as comorbidities and temporal skew of creatinine play a more important role in model predictions. A deeper discussion on the feature importance follows below.

2. *Additionally, the practical utility of the created model for physicians at the bedside in the ICU remains unclear. Developing an algorithm (similar to the kidney failure risk equation), where variables can be inputted to estimate the risk of death over the subsequent 7 days, would enhance its usefulness. A risk of death > 90% can be considered futile.*

Response: Our model represents the first “base model” of this type and will be assessed for clinical utility in following work. Our current work demonstrates that a tool for recommending CRRT is a potentially solvable problem, and that our data contains enough information to help inform a decision.

3. *Currently, the clinical relevance of MCV measurements and tobramycin levels, as described, is uncertain.*

Response: As discussed in a previous response, the top 10 most important features to the model only partially

illustrate what the model is basing its decisions on. This feature list is unable to address the importance of the complex interactions between subsets of the hundreds of features in the EHR the model also utilizes. As discussed previously, surgery before CRRT is a top variable that predicts mortality and is consistent with published literature (PMID: 28636702; 31711002). Other features including white blood cell count and creatinine, have been linked to excess mortality (PMID: 28636702; 32028984).

4. *The article, particularly in the discussion section, could benefit from a simpler presentation. It should include commentary on similar predictive models from prior research and explicitly highlight the distinctions from other models.*

Response: We thank the reviewer for this point and have simplified the presentation to the extent possible. In addition, we included comparison with previous predictive models and discuss how our model is unique and differs (lines 102-104 in the Introduction section; lines 526-531 in the Discussion section).

Reviewer 3

1. *The authors created a matched population from the same hospital making it a single case control cohort with UCLA-CRRT as cases and UCLA-matched as controls. They matched patients according to demographics and clinical characteristics. This is similar to propensity score matching to reduce selection bias, but unfortunately treatment decision is very likely not random (treatment bias still exists). Can the authors define in this matched population why some patients were treated with CRRT and some are not? Comorbidity might be an explanatory factor since this is higher in the "not recommended" cases vs controls.*

Response: We thank the reviewer for the comment. As discussed, in lines 656-659, we describe the control cohort and explain that none of them are placed on CRRT. The purpose of this cohort was to capture patients who did not go on CRRT but were similar to the CRRT populations based on age, sex, race, ethnicity, and Charlson Comorbidity Index.

2. *line 680, it is not clear what the authors mean with patients should not be placed on CRRT and are labeled as such: isn't this a retrospective study where the patients are selected as controls as such?*

Response: We agree with the reviewer that our description for generating labels for the control population was unclear. As discussed, none of the control population were placed on CRRT. We label each control patient as not recommended for CRRT. We have amended lines 680-681 to be clearer: "All patients in the UCLA: Control cohort were not placed on CRRT and therefore did not have CRRT outcome labels. We therefore assigned these patients a negative outcome (to not recommend CRRT) based on the observation that they were not placed on CRRT."

3. *Line 703: Based on what did the clinicians approve the feature list? Which criteria were used?*

Response: The features were solely determined by the model after analyzing the EHR records. There was no clinician involvement. We removed this line from the manuscript.

4. *Line 820: Were features missing completely at random or were there dependencies? How did this influence imputation?*

Response: We thank the reviewer for the comment. It is possible that we can test each individual feature for MCAR and MCAR only. It would, however, be impossible to tell if there were any interactive or multivariate mechanisms occurring. It is also not clear in our view how knowing which features were MCAR on the univariate level would be useful. We have tried simple imputation (which would introduce bias) and KNN (that could also possibly introduce bias by eliminating variability). These are valuable approaches because they are different categories of imputation procedures. However, we subsequently removed KNN in the most recent iteration as we observed it was considerably slow and did not outperform simple imputation. We could expand to MICE, but anything beyond that we deemed to be overly complex and be associated with an additional cost. MICE is rather slow, more-so than KNN, but guarantees unbiased estimates under MAR. We similarly observed that it did not make that much of a difference to overall performance. It is interesting to note, more accurate

imputation does not always mean better performance for prediction.

5. *Precision (in PRAUC) is dependent on prevalence (positive predictive value): were all patients in the respective hospitals included for analysis in the predefined time periods, i.e. is the prevalence correct?*

Response: To our knowledge, our dataset consists of all available patients at these institutions that were recorded into the EHR.

6. *How fair were the designed algorithms? How were race and ethnicity defined? It seems from fig A3 that the algorithms are somewhat underperforming for some of the minority groups.*

Response: We have characterized predictive imbalances based on common demographical and clinical subgroups in our analysis, such as Figure 5A in the Results Section, and Section 4.4 Subpopulation Analysis in the Methods section. We present the groups based on race and ethnicity in our tables and graphs. For example, Figure 5A shows data on Asian, Black or African American, Unknown, and White or Caucasian groups. Ethnicity is broken down into Hispanic or Latino, Not Hispanic or Latino. In our current iteration, we detected less underperformance in these groups. These labels are drawn from self-reported demographics captured within each institution's EHR.

7. *There also seems to be worse performance in older patients.*

Response: We do see slightly worse performance in older patients. We believe that this may be due to the sample size of this group. The text has been amended: "We observed a slightly poorer and greater variation of performance ... patients 90 to 100 years old (N = 67 (2.0%)), which is tied to the sample size of these groups."

8. *Four machine learning models were chosen, but it is unclear why these were selected, could the authors elaborate on this?*

Response: We thank the reviewer for the question. These four are very commonly used machine learning models (mix of linear and nonlinear) that have been shown to give the highest performance across many domains and are structured in a way that allows the inspection of the models for decision-making transparency. To make this clearer, we have amended the text under Methods 4.3.2: "Candidate models m included linear and non-linear models: light gradient-boosting machine LGBM), extreme gradient-boosted decision tree (XGB), random forest (RF), or logistic regression (LR)."

9. *Fig 2: what would be the effect of the overestimation at high probability and underestimation at low probability? Can the authors recalibrate perhaps?*

Response: With the updated results, our model is well-calibrated, so we do not need to recalibrate the model. Overestimating at high probability means that the model will be overconfident when recommending CRRT, and underestimating at low probability means the model will be unsure about not recommending CRRT. If we needed to calibrate the model, we could use Platt scaling or isotonic regression.

10. *Perhaps I misread, but did the authors perform ensembling of the 4 machine learning models as well? or majority voting perhaps?*

Response: We thank the reviewer for this comment. We did not perform ensembling, however, we agree it is an interesting idea that we intend to explore in our future work.

11. *Could the authors perform a net benefit analysis (decision curve analysis)? for section 2.4, relatively similar to fig 5b*

Response: We agree with this insight. We have replaced Figure 5b with a net benefit analysis.

12. *I'm not sure if fig 5b shows the right clinical scenario for potential implementation, because the fact that a patient is treated with CRRT is dependent on a previous choice, which suffers from treatment bias. What about patients not being placed on CRRT, but who would benefit? How does the 43% CRRT+poor outcome compare to other hospitals for instance? What would be the effect on the net benefit curve when this prevalence is changing (up or down)? Could the authors perhaps perform a simulation?*

Response: As suggested, we replaced Figure 5b with a net benefit analysis. Our model (as all models of this kind that deal with patients' outcomes) will be ultimately tested in future prospective clinical trial to assess clinical utility (lines 621-624).

13. *The decrease in model performance upon external evaluation is quite extensive: do the authors have an explanation (and perhaps also mitigation strategy) for this domain shift?*

Response: We thank the reviewer for this comment. The difference in model performance is likely due to the discrepancy in data collection practices across our two cohort datasets. For example, there are significantly more labs for the UCLA dataset than Cedars, whereas Cedars has more vital signs in their EHR. While we acknowledge this difference, it aligns with the need to fine tune these types of models at each institution that will potentially use it clinically (lines 563-569). Additionally, we show in Figure 2 and A2 (in the updated graphs) that training across both institutions produces better results than training at a single institution and applying it to another. We have added the following text after line 475: "These results suggest that the model learned meaningful relationships in the data before starting CRRT that maintained importance in the UCLA: CRRT cohort but not in the Cedars: CRRT cohort... We hypothesize that the decrease was due to relatively higher number of labs available in the UCLA: CRRT cohort compared to the Cedars: CRRT cohort..." We also add the following to the Discussion: "...we also show that local solutions may be adequate or even better than general solutions. However, this is only true if local solutions are deployed in appropriate contexts, especially if resources are constrained (e.g., limited access to training and evaluation data). Particularly, one aspect of an inappropriate localization would be to directly transfer another local model to a different institution. We observe decreases in performance due to domain shift which is potentially due to discrepancies in data collection practices across institutions (e.g., higher number of labs available in the UCLA: CRRT cohort compared to the Cedars: CRRT cohort)."

14. *At the end, the model is for decision support, which indicates that the physician (+ patient) make the decision to initiate treatment and to continue or not. It might be worthwhile to perform a small crossover trial where physicians are randomized in either or not supported by the tool to see the impact on decision making.*

Response: We thank the reviewer for this comment. A future clinical trial will be essential to document the utility of the model clinically compared to physician-based decision making. We added the following text: "Another informative case analysis would compare decision-making with and without model assistance."

REVIEWER COMMENTS

Reviewer #1 (Remarks to the Author):

I would like to thank the authors and acknowledge the significant effort that was put in to incorporating feedback for this second draft. The manuscript is well written and much clearer. The model is innovative and provides a novel method for assessing the benefit of CRRT for individual patients. The authors have addressed the previous comments. I have two additional minor suggestions below.

1. Effect size

There is a description of what analyses were used to calculate effect size and the results are represented in Figs 4 and A8. However, there was no description in the text as to how was used and how they should be interpreted. It would be helpful if this was clarified.

2. Clarification of CRRT assignment in the original datasets.

It would be useful if the authors stated that the patients were already categorized into CRRT or no CRRT groups prior to receiving the original datasets. This would ensure that readers do not think CRRT was assigned based on ICD9/10 codes. The issue with this is that there is only one hemodialysis ICD9 code that refers to intermittent, CRRT and hybrid modalities. This code does not convert to ICD10 codes where there is a specific code for each hemodialysis type.

Reviewer #3 (Remarks to the Author):

The authors have addressed most of the questions sufficiently, but I have some points of discussion left. Below the pointwise comments (corresponding to the numbering in the rebuttal):

R3Q1: the quality of matching is important for their model and I understand that none of the patients were (per definition) placed in CRRT. Matching seems to give balance in the mentioned (measured) characteristics age, sex, race, ethnicity and CCI, but this leaves the

question why these patients were placed on CRRT and the controls were not. I would like to refer to the article by Fu et al. 2019 Nephrol Dial Transplant, where the authors discuss the caveats of using propensity score matching. It would therefore be very worthwhile for the reader to understand what has led to the decision to start CRRT and I would like to invite the authors to address this issue in more detail (e.g. identifying factors that have led to this choice, i.e. other characteristics).

R3Q4: addressing imputation with MICE indeed gives more balanced imputation and it can nicely address uncertainty related to the imputation procedure by pooling the estimates of the various M datasets. I would advise the authors indeed to perform MICE and create pooled estimates of their machine learning approach (i.e. running the machine learning models on each of the M datasets followed by estimate and variance pooling with e.g. Rubin's rule. MNAR compared to MCAR can give flawed estimates, also on the univariate level. I agree with the authors that more accurate imputation does not always lead to better performance for prediction, but this is also not the goal of imputation. Imputation should give a potentially better estimate of the true performance of the prediction model (which can be better or worse in performance, depending on the underlying true data structure).

Reviewer #3 (Remarks on code availability):

I did not try to run the code myself, but the installation guide on github and the organization of the code seems adequate. I do not foresee any installation problems as all python packages are readily available in conda/pip and the conda environment with version control is defined in a separate yaml file. README documentation is adequate, although in-code annotation could be a bit extended

Reviewer #4 (Remarks to the Author):

I appreciate the authors' efforts to respond to the reviewers' comments and revise the manuscript accordingly. However, several critical issues as below still need to be addressed.

1. What is the primary purpose of this model? The model needs to be revised by implementing more detailed clinical variables for improved translational value. The sophisticated model per se may be statistically acceptable, however clinicians need a more

practical and intuitive model that can be applied in real world practice.

2. Ambiguity of feature importance should be resolved. For example, the type of surgery and infection needs to be stratified according to the degree of severity. The skew of several important laboratory values such as serum creatinine, bicarbonate, ALP, hemoglobin, MCV, and WBC need to be defined and a threshold or cutoff value should be presented.

3. Baseline characteristics of all cohorts should be presented. This is one of many basic requirements of medical papers.

4. Reanalysis of the data leading to development of a more practical and simplified model according to the types of ICUs (MICU, CCU, or SICU) will substantially improve the translational robustness of the study.

5. Please correct grammatical errors including inappropriate capitalizations such as 'Recover Renal Function, Transition to Hemodialysis (Inpatient), Comfort Care, and Expired' on page 18.

Below, we respond to the comments provided (please note that the line numbers referred to in this document are in relation to the original submission). Once more, we believe our responses and these changes address all the concerns raised in the review and significantly strengthen the overall impact of our work. We look forward to your consideration of this revised manuscript.

Reviewer 1

1. *Effect size. There is a description of what analyses were used to calculate effect size and the results are represented in Figs 4 and A8. However, there was no description in the text as to how was used and how they should be interpreted. It would be helpful if this was clarified.*

Response: We have updated the text to add the following after line 980: “*Effect sizes complement the significant results to provide a measure of strength of difference, and thus highlight the features with values that differ the most between incorrectly classified samples and their correct counterparts...If we rejected, we reasoned that the model may have been using the feature to incorrectly distinguish the populations due to factors such as confounding, and such features with the largest effect sizes should be considered for additional analyses.*”

2. *Clarification of CRRT assignment in the original datasets. It would be useful if the authors stated that the patients were already categorized into CRRT or no CRRT groups prior to receiving the original datasets. This would ensure that readers do not think CRRT was assigned based on ICD9/10 codes. The issue with this is that there is only one hemodialysis ICD9 code that refers to intermittent, CRRT and hybrid modalities. This code does not convert to ICD10 codes where there is a specific code for each hemodialysis type.*

Response: We have now added the following clarification at line 662: “*CRRT patients were already identified prior to receiving the datasets, and therefore no ICD codes were used to categorize patients into the CRRT and UCLA: Control cohorts.*”

Reviewer 3

1. *Initial question (previous revision): The authors created a matched population from the same hospital making it a single case control cohort with UCLA-CRRT as cases and UCLA-matched as controls. They matched patients according to demographics and clinical characteristics. This is similar to propensity score matching to reduce selection bias, but unfortunately treatment decision is very likely not random (treatment bias still exists). Can the authors define in this matched population why some patients were treated with CRRT and some are not? Comorbidity might be an explanatory factor since this is higher in the "not recommended" cases vs controls.*

Initial response: As discussed, in lines 656-659, we describe the control cohort and explain that none of them are placed on CRRT. The purpose of this cohort was to capture patients who did not go on CRRT but were like the CRRT populations based on age, sex, race, ethnicity, and Charlson Comorbidity Index (CCI).

Follow-up question: The quality of matching is important for their model and I understand that none of the patients were (per definition) placed in CRRT. Matching seems to give balance in the mention (measured) characteristics age, sex, race, ethnicity and CCI, but this leaves the question why these patients were placed on CRRT and the controls were not. I would like to refer to the article by Fu et al. 2019 Nephrol Dial Transplant, where the authors discuss the caveats of using propensity score matching. It would therefore be very worthwhile for the reader to understand what has led to the decision to start CRRT and I would like to invite the authors to address this issue in more detail (e.g. identifying factors that have led to this choice, i.e. other characteristics).

Follow-up response: The objective of this study was to evaluate all available variables to better understand why some CRRT patients benefited despite the decision to put them all on CRRT. The primary focus of model evaluation was on patients who underwent CRRT; in other words, we were not comparing patients on CRRT as an intervention group against persons not on CRRT; we were instead identifying models for predicting short-term survival in all patients placed on CRRT. Our approach of including a general non-CRRT control population

with similar underlying sickness was to contribute to the model generation phase and enhance the scrutiny of the model by identifying patterns that may be more specific to patients requiring renal replacement therapy. For instance, excluding non-CRRT patients could bias the discrimination of those who would benefit from CRRT to those who were healthier in the CRRT population; on the other hand, given non-CRRT patients of similar disease status, the model, through a data-driven approach, should differentiate those that would benefit from CRRT from those that would not. The improved performance when including the control cohort suggests the identification of patterns that inform different decisions among those who were placed on CRRT and the controls. Furthermore, the feature analysis provides insight into the identifying factors that contribute to the choice of whether patients should be placed on CRRT (as opposed to a knowledge-driven approach). Given the retrospective limitations of the data, we do not know whether some of the control patients were considered for CRRT as such information is typically not documented; the implicit assumption that prevents changing the data was that not placing them on CRRT was the correct decision. We recognize the possibility of the control patient potentially benefiting had they been placed on CRRT, which is a consequence of the challenge that inspired this study, i.e., the difficulty identifying those who would benefit from CRRT. To this end, we add in lines 621-623 that we will perform an in-depth case-by-case analysis of the control cohort in future work.

In general, we recognize the issues of using propensity scores as a matching criterion when treated patients are being compared with untreated patients, though in a retrospective framework, this strategy at least provides some context for comparison. Overall, the issues are also not critical as the focus of model evaluation was on patients who received CRRT.

2. *Initial question (previous revision): Line 820: Were features missing completely at random or were there dependencies? How did this influence imputation?*

Initial response: We thank the reviewer for the comment. It is possible that we can test each individual feature for MCAR and MCAR only. It would, however, be impossible to tell if there were any interactive or multivariate mechanisms occurring. It is also not clear in our view how knowing which features were MCAR on the univariate level would be useful. We have tried simple imputation (which would introduce bias) and KNN (that could also possibly introduce bias by eliminating variability). These are valuable approaches because they are different categories of imputation procedures. However, we subsequently removed KNN in the most recent iteration as we observed it was considerably slow and did not outperform simple imputation. We could expand to MICE, but anything beyond that we deemed to be overly complex and be associated with an additional cost. MICE is rather slow, more-so than KNN, but guarantees unbiased estimates under MAR. We similarly observed that it did not make that much of a difference to overall performance. It is interesting to note, more accurate imputation does not always mean better performance for prediction.

Follow-up question: Addressing imputation with MICE indeed gives more balanced imputation and it can nicely address uncertainty related to the imputation procedure by pooling the estimates of the various M datasets. I would advise the authors indeed to perform MICE and create pooled estimates of their machine learning approach (i.e. running the machine learning models on each of the M datasets followed by estimate and variance pooling with e.g. Rubin's rule. MNAR compared to MCAR can give flawed estimates, also on the univariate level. I agree with the authors that more accurate imputation does not always lead to better performance for prediction, but this is also not the goal of imputation. Imputation should give a potentially better estimate of the true performance of the prediction model (which can be better or worse in performance, depending on the underlying true data structure).

Follow-up response: We thank the reviewer for the discussion. MICE, of course, is well-appreciated and guarantees unbiased estimates under MAR. We tried MICE on a limited subset of the data but given the number of features considered in the model, this was not tractable; furthermore, on the dataset that we applied MICE, its performance was suboptimal compared to the simpler method we used (i.e., simple imputation). We agree that if the goal was to predict the missing values, then a more faithful imputation method such as MICE may be suitable and require a separate study to investigate. However, our goal was to predict the outcome of success after being treated with CRRT. Given that we will never know the actual values of the missing information, we view the goal of imputation in our context as an intermediate step towards prediction, regardless of accuracy of

imputation itself. Markedly, in other works we have noted that accurate imputation does not always mean better performance for data-driven prediction, and the choice of “optimal” imputation method is highly dependent on the predictive task and domain – there is no “singular” approach to imputation that guarantees optimal performance.

It is also not certain that MICE will lead to a truer performance of the model. And as mentioned above, we tested the pipeline with MICE and observed a decrease in predictive performance. As the actual values will remain unknown, we cannot evaluate the impact of imputation techniques relative to performance had the missing data been observed. Furthermore, as we do not know what missingness mechanisms are occurring in the data in both a univariate and multivariate case, we can conversely theorize that we are inferring the most information possible (without imparting additional assumptions) when using simpler imputation methods. The simple methods introduce uniformity across missing values and implicitly encode the fact that the data was missing as information itself, which may be more reflective of real-life application. Still, neither claim about MICE nor simple imputation can be substantiated directly in this situation; thus, we assert that the goal of imputation should be to improve the downstream predictive task and should be evaluated within such a context.

3. *I did not try to run the code myself, but the installation guide on github and the organization of the code seems adequate. I do not foresee any installation problems as all python packages are readily available in conda/pip and the conda environment with version control is defined in a separate yaml file. README documentation is adequate, although in-code annotation could be a bit extended.*

Response: We thank the reviewer for the comment. We have extended the in-code annotations and will maintain good coding practices.

Reviewer 4 (New)

1. *What is the primary purpose of this model? The model needs to be revised by implementing more detailed clinical variables for improved translational value. The sophisticated model per se may be statistically acceptable, however clinicians need a more practical and intuitive model that can be applied in real world practice.*

Response: The overarching intent of this study was to assess if a data-driven approach could identify any meaningful patterns from CRRT data, determining if: 1) a comprehensive assessment of the data could uncover any “signal” that would inform predictions; and 2) a framework for baseline assessment of individuals prior to commencing CRRT would be useful. Both objectives are demonstrated in this effort. We acknowledge the challenge of interpreting the model presented in this paper and the desire for a more intuitive model that can be applied in real-world practice, moving towards clinical translation. Our main objective, however, was not to develop a translational model that could be immediately deployed clinically, but to demonstrate the utility of machine learning with a novel dataset for CRRT and to then identify a space of features to consider. Indeed, our next steps will bring the model closer to clinical translation through development of a parsimonious model with improved interpretability and applicability. In lines 630-636, we agree that, “...simpler versions of this predictive model could make the model more easily understood by clinicians and more accessible to a wider range of institutions.” As part of the previous revision, we also included the following in the introduction: “*Though the model as presented is not currently intended for clinical use, our work demonstrates its utility as incentive for clinical trials in the future.*”

Ultimately, we believe that this work demonstrates merit in that there is useful “signal” in the data and that a parsimonious model may be possible in this domain; however, such translation may also be highly site-specific, which we will assess in a future study involving a broader number of institutions.

2. *Ambiguity of feature importance should be resolved. For example, the type of surgery and infection needs to be stratified according to the degree of severity. The skew of several important laboratory values such as serum creatinine, bicarbonate, ALP, hemoglobin, MCV, and WBC need to be defined and a threshold or cutoff value should be presented.*

Response: Stratifying according to the type of surgery and infection severity would be invaluable for clinical

translation. But if we were to do so, there would not be sufficient observations for the analysis. Our objective was to identify “signal” differentiating patients who did and did not benefit from CRRT. Stratification will be incorporated as we continue future data collection and work towards a simpler (parsimonious) model, appreciating that the datasets are potentially rich enough for such analyses.

Rather than pre-specifying variables such as degree of severity of sepsis or type of surgery, we created the model based on all available variables in the EHR. Importantly, severity of sepsis is captured by other variables such as lactate values and the use of pressors. The lab values included in the model are studied as continuous variables and therefore there are no specific cut-offs or thresholds that were prespecified. Furthermore, the model uses different levels for individual lab value variables based on interaction with levels of other variables. For example, the strength of white cell counts in the model can be in part dependent on values of other related variables such as lactate levels.

3. *Baseline characteristics of all cohorts should be presented. This is one of many basic requirements of medical papers.*

Response: A table of baseline characteristics for all cohorts is provided in the supplemental materials as part of Table A1.

4. *Reanalysis of the data leading to development of a more practical and simplified model according to the types of ICUs (MICU, CCU, or SICU) will substantially improve the translational robustness of the study.*

Response: We appreciate this comment. As requested, we have completed an additional subgroup analysis and now include performance metrics in MICU, SICU, or CCU patients (see supplement Table A3). We also add the following at line 578 in the discussion: “*We demonstrate a preliminary extension of the analysis to different ICU types (Extended Data Table A3), which reveals strong model performance on surgical ICU patients, while generally worse performance on medical ICU patients especially from the Cedars: CRRT cohort. Such findings present crucial considerations as we improve the models for robust translation.*” Some of the other ICUs such as neuro ICUs could not be done due to limitations in sample size.

Yet per Q2, the consideration of *separate models* for different ICU settings would be more appropriate as we work towards a simpler model for clinical translation. The scope of this work was to highlight the utility of machine learning with a novel CRRT dataset and identify a large feature-space that we can refine in future steps to build a parsimonious model with improved translatability.

5. *Please correct grammatical errors including inappropriate capitalizations such as ‘Recover Renal Function, Transition to Hemodialysis (Inpatient), Comfort Care, and Expired’ on page 18.*

Response: These changes have been corrected as requested.

REVIEWERS' COMMENTS

Reviewer #3 (Remarks to the Author):

No further questions

Reviewer #3 (Remarks on code availability):

see earlier response

Reviewer #4 (Remarks to the Author):

The authors addressed all issues that I raised and improved the quality of the manuscript.

I have no additional comment.